# VTA-projecting cerebellar neurons mediate stress-dependent depression-like behaviors

**Soo Ji Baek[1,2], Jin Sung Park[1,2], Jinhyun Kim[1,2], Yukio Yamamoto[1], Keiko Tanaka-Yamamoto[1,2]***

[1]Center for Functional Connectomics, Brain Science Institute, Korea Institute of Science and Technology (KIST), Seoul, Republic of Korea; [2]Division of Bio-Medical Science and Technology, KIST School, Korea University of Science and Technology (UST), Seoul, Republic of Korea

**Abstract** Although cerebellar alterations have been implicated in stress symptoms, the exact contribution of the cerebellum to stress symptoms remains to be elucidated. Here, we demonstrated the crucial role of cerebellar neurons projecting to the ventral tegmental area (VTA) in the development of chronic stress-induced behavioral alterations in mice. Chronic chemogenetic activation of inhibitory Purkinje cells in crus I suppressed c-Fos expression in the DN and an increase in immobility in the tail suspension test or forced swimming test, which were triggered by chronic stress application. The combination of adeno-associated virus-based circuit mapping and electrophysiological recording identified network connections from crus I to the VTA via the dentate nucleus (DN) of the deep cerebellar nuclei. Furthermore, chronic inhibition of specific neurons in the DN that project to the VTA prevented stressed mice from showing such depression-like behavior, whereas chronic activation of these neurons alone triggered behavioral changes that were comparable with the depression-like behaviors triggered by chronic stress application. Our results indicate that the VTA-projecting cerebellar neurons proactively regulate the development of depression-like behavior, raising the possibility that cerebellum may be an effective target for the prevention of depressive disorders in human.

*For correspondence:
keikoyamat@gmail.com

**Competing interest:** The authors declare that no competing interests exist.

## Editor's evaluation

This study provided novel insight into a role of a cerebellar-ventral tegmental area (VTA) circuit that is recruited by chronic stress exposure and can influence affective behavior. A combination of experimental manipulations, carefully performed and analyzed, dissecting the cerebello-VTA circuit with neuroanatomy, c-fos expression, behavior, and chemogenetics, supports the conclusion of the paper, that the VTA-projecting cerebellar neurons proactively regulate the development of stress-dependent changes in affective behavior. This report provides convergent evidence for a novel pathway that modulates chronic stress-related behaviors, and makes an important advance in the field.

## Introduction

Whereas the cerebellum has traditionally been considered to be important solely for motor coordination and learning, it became apparent that the cerebellum is also involved in higher cognitive functions (*Rochefort et al., 2013*; *Hull, 2020*; *Wagner and Luo, 2020*). One such non-motor cognitive function is reward-related processing (*Carta et al., 2019*; *Medina, 2019*; *Hull, 2020*; *Sendhilnathan et al.,*

*2020*). Consistent with the common notion that reward circuitry regulates stress-driven behavioral changes or depressive symptoms (*Nestler and Carlezon, 2006*; *Russo and Nestler, 2013*; *Pignatelli and Bonci, 2015*; *Fox and Lobo, 2019*), many studies have demonstrated that abnormalities of the cerebellum are associated with stress responses or mental disorders, latter of which include major depressive disorder and post-traumatic stress disorder that can be triggered or worsened by stress. The cerebellum is affected by stress, as is evident from the alterations in function, structure, and molecular expression that occur in the cerebellum after the exposure of animals or humans to stressful events (*Gounko et al., 2013*; *Huguet et al., 2017*; *Bambico et al., 2018*; *Moreno-Rius, 2019*). Alterations of the cerebellum, such as decreased volume, abnormal neuronal activity, and disrupted cortical connectivity, have also been observed in patients with major depressive disorder and post-traumatic stress disorder, and such alterations are often correlated with symptoms of these disorders (*De Bellis and Kuchibhatla, 2006*; *Alalade et al., 2011*; *Baldaçara et al., 2011*; *Liu et al., 2012*; *Guo et al., 2012*; *Guo et al., 2013*; *Phillips et al., 2015*; *Córdova-palomera, 2016*; *Xu et al., 2017*; *Depping et al., 2018*; *Moreno-Rius, 2019*). Although the accumulating lines of evidence indicate the involvement of the cerebellum in stress responses and stress-associated disorders, the exact role played by the cerebellum remains unclear.

The cerebellum interacts with many brain areas through both direct and indirect synaptic connections, and one of the brain areas receiving direct inputs from the deep cerebellar nuclei (DCN), which is the major cerebellar output, is the ventral tegmental area (VTA) (*Snider et al., 1976*; *Parker et al., 2014*; *Beier et al., 2015*; *Carta et al., 2019*). The VTA is the origin of dopaminergic neurons projecting to other reward-related brain regions, so that it is considered as a central component of reward circuitry. The VTA is also known to be involved in the regulation of stress susceptibility (*Fox and Lobo, 2019*). Stressful events alter VTA dopamine (DA) neuron activities, and stress susceptible phenotypes are linked to the modulation of VTA DA neuron activity (*Chaudhury et al., 2013*; *Tye et al., 2013*; *Friedman et al., 2014*; *Isingrini et al., 2016*). The VTA receives inputs from many brain regions (*Beier et al., 2015*; *Watabe-Uchida et al., 2012*; *Zahm et al., 2011*). Some of these inputs may determine stress-associated responses via the regulation of VTA DA neuron activities (*Lammel et al., 2012*; *Isingrini et al., 2016*; *Fernandez et al., 2018*; *Knowland et al., 2017*). However, it remains to be elucidated as to how environmental or physical factors that affect the development of stress symptoms, process neural network mechanisms underlying a variety of neuronal activities in the VTA and stress responses. The VTA is generally considered to be a minor target of DCN neurons, yet its projections were reported to be functionally sufficient as their optogenetic stimulation robustly increased the activity of VTA neurons (*Carta et al., 2019*). In addition, the DCN projection to the VTA was shown to be associated with reward (*Carta et al., 2019*). Considering that the circuitry underlying reward is often closely associated with the stress susceptibility and resilience, the idea of active cerebellar involvement in the development of stress symptoms, presumably through the pathway to the VTA, appears reasonable.

In this study, we utilized chemogenetic manipulation during chronic stress application, and found that activation of cerebellar Purkinje cells (PCs) in crus I prevented the stress-induced behavioral changes. We further identified a neuronal circuit from crus I of the cerebellar hemisphere to the VTA through the dentate nucleus (DN) of the DCN, and demonstrated that DCN neurons wiring this pathway are indeed involved in the development of behavioral changes. We thus propose a possibility that cerebellar functionality contributes to the cause of variabilities in stress susceptibility.

## Results

### The cerebellum is involved in the development of stress-dependent behavioral changes

We first set up an experimental procedure of chronic restraint stress (RS) application and a subsequent test to evaluate stress-mediated behavioral changes, using a similar protocol to that established in previous studies (*Kim and Han, 2006*; *Park et al., 2010*; *Kim and Leem, 2014*; *Kim et al., 2015b*; *Son et al., 2019*). Wild-type mice were exposed to RS for 2 hr daily for 2 weeks (*Figure 1A*, RS group), whereas mice in the control (CTR) group were in their home cages in the animal facility without RS exposure. During the following 3 days, behavioral tests were performed in the order of open-field test (OFT), tail suspension test (TST), and forced swimming test (FST) (*Figure 1A*). The TST and FST were

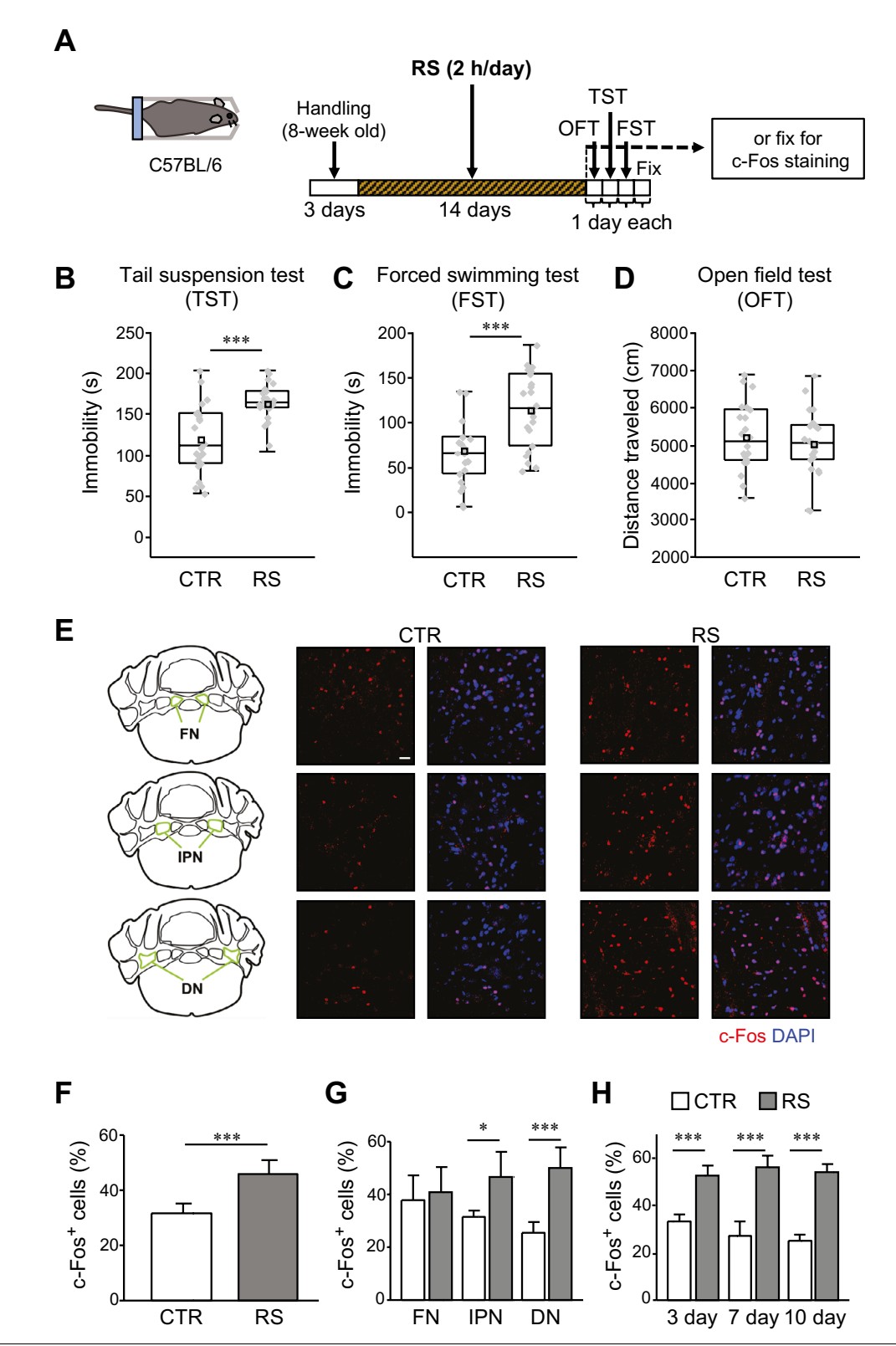

**Figure 1.** c-Fos expression in the DCN is increased by the chronic stress application. (**A**) Schematic diagram of the RS (2 hr/day) application for 2 weeks, followed by 1 day each for the OFT, TST, and FST before fixation, or by fixation for c-Fos staining. (**B–D**) Immobility time in the TST (**B**) and FST (**C**), and total distance moved in the OFT (**D**), to analyze the effects of RS in C57BL/6 wild type mice. ***p < 0.001, Student's *t*-test (n = 22 mice each

*Figure 1 continued on next page*

*Figure 1 continued*

for the CTR and RS groups). Behavioral data in this and subsequent figures are presented as boxplots with gray dots representing individual data points, center lines denoting the median, open square dots denoting the mean values, the lower and the upper bounds of the box corresponding to the 25th and 75th percentiles, respectively, and the whiskers denoting the minimum and maximum values. Effect sizes for behavioral data are available in *Figure 1—source data 2*. (**E**) 3D projection images of c-Fos expression and DAPI staining in the FN (top), the IPN (middle), and the DN (bottom) of the CTR (middle) and RS (right) groups. The overview diagrams on the left indicate the location of three nuclei. Scale bar: 20 µm. (**F–H**) Comparisons of the percentage of c-Fos-positive (c-Fos[+]) cells out of DAPI-positive cells between the CTR and RS groups in the whole DCN (**F**) or each nucleus (**G**) examined after 2-week RS application, or in the DN examined after 3-, 7-, and 10-day RS application (**H**). *p < 0.05, ***p < 0.001, Student's *t*-test comparing between CTR and RS groups, (**F** and **G**), n = 8 images each in 3 nuclei of the CTR or RS group, from 4 mice each in the CTR or RS group; (**H**) n = 5 images each in the DN of the CTR or RS group, from 5 mice each in the CTR or RS group. Data are shown as the mean ± SEM. Exact p values and the statistical tests used are available in *Figure 1—source data 1*.

The online version of this article includes the following source data and figure supplement(s) for figure 1:

**Source data 1.** p Values and statistical tests related to *Figure 1*.

**Source data 2.** Effect sizes for behavior results related to *Figure 1*.

**Figure supplement 1.** Conditioning of mice for the RS protocol.

**Figure supplement 2.** Cell types for c-Fos-positive DCN neurons in the DN following 14 days RS.

used to assess stress-mediated behavioral changes, unless otherwise stated, and the OFT was used to examine general locomotor activity. The behavioral analyses demonstrated that the RS group had increased immobility in both the TST (*Figure 1B*, *Videos 1 and 2*; p < 0.001) and FST (*Figure 1C*, *Videos 3 and 4*; p < 0.001) compared with the CTR group. The increase in immobility is unlikely to be due to the reduction of general locomotor activity, because there was no significant difference in travel distance measured by the OFT between the CTR and the RS groups (*Figure 1D*, *Figure 1—figure supplement 1A*; p = 0.52). We also measured the time spent in the center or peripheral areas in the OFT. Although animals receiving chronic stress has sometimes shown the decreased time spent in the center (*Kim and Han, 2006*; *Choi et al., 2014*; *Simard et al., 2018*; *Song et al., 2020*), our results showed no significant difference in the time between the CTR and the RS groups (*Figure 1—figure supplement 1B*; p = 0.09). Through these behavioral tests, we confirmed that chronic mild stress by the application of 2 hr of daily RS for 2 weeks triggers behavioral changes assessed by the TST and FST in mice. The increase in immobility in the TST and FST has been considered as a despair-like behaviors (*Wang et al., 2017*; *Planchez et al., 2019*), which is similar to one of phenotypes observed in patients with depression. We refer to the stress-mediated behavioral changes observed in this study as depression-like behaviors that have been used to express behavioral changes triggered by chronic stress in rodent models of depression (*Wang et al., 2017*; *Gururajan et al., 2019*; *Planchez et al., 2019*).

A previous study showed that application of another type of chronic mild stress for 3 weeks enhanced the expression of an immediate early gene, c-Fos, in the DCN (*Huguet et al., 2017*), indicating an increase in the activity of DCN neurons. To clarify whether the depression-like behaviors induced by our RS application protocol are associated with cerebellar activity, we compared c-Fos expression in the DCN between the CTR and the RS groups. We separated the DCN into subregions,

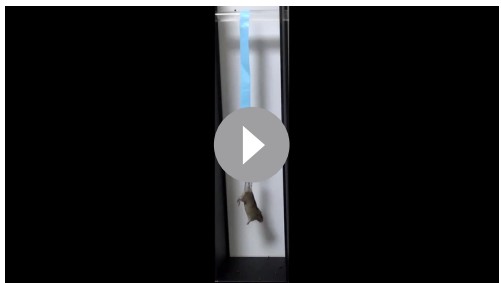

**Video 1.** TST of the CTR group.
https://elifesciences.org/articles/72981/figures#video1

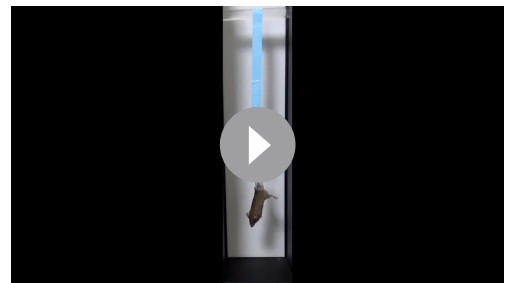

**Video 2.** TST of the RS group.
https://elifesciences.org/articles/72981/figures#video2

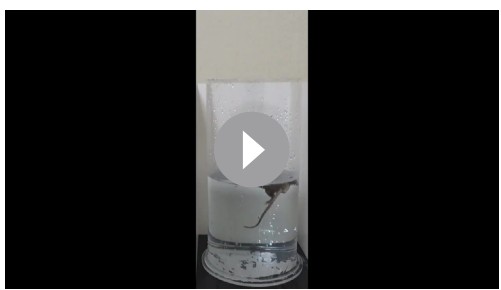

**Video 3.** FST of the CTR group.
https://elifesciences.org/articles/72981/figures#video3

that is the fastigial nucleus (FN), the interposed nucleus (IPN), and the DN, to analyze possible differences in c-Fos expression in the various functional regions (*Figure 1E*, *Figure 1—figure supplement 1C and D*). The expression of c-Fos in the percentage of all DAPI positive cells for the RS group was significantly enhanced (*Figure 1F*; p < 0.001), particularly in the DN and in the IPN (*Figure 1G*; DN: p < 0.001; IPN: p = 0.036), where notably more c-Fos-positive neurons were observed by immunohistochemical analysis of cerebellar slices that were fixed immediately after the end of the 2 week RS application. In contrast, there was no substantial change in c-Fos level in the FN (*Figure 1G*; p = 0.74). This data of the changes in c-Fos expression indicates the activation of DCN neurons in the IPN and the DN by chronic mild stress. In addition, we tested the cell types of these c-Fos-positive cells by co-staining with either NeuN, GAD67, or GFAP. NeuN and GAD67 are markers of neurons and inhibitory neurons, respectively, that were also used to stain the DCN neurons (*Molineux et al., 2006*; *Lee et al., 2015*). Most of the c-Fos-positive cells had an overlap with NeuN (*Figure 1—figure supplement 2A*), but not with GAD67 (*Figure 1—figure supplement 2B*). As previously reported (*Groteklaes et al., 2020*), clear signals of GFAP, a marker of glial cells, were not detected in the DCN, and consequently, cFos-positive cells in the DCN were GFAP negative (*Figure 1—figure supplement 2C*). These results indicate that cells in the DCN activated by RS are glutamatergic neurons. To further examine whether the activation of DCN neurons is a consequence observed only after the chronic RS application or an event that happens during the process of the RS, the c-Fos expression in the DN, which showed the greater increase than the IPN, was analyzed after 3-, 7-, or 10-day RS application. The c-Fos expression in the RS group was significantly enhanced compared with the CTR group even by 3 days of RS application, and was maintained at elevated level after 7- or 10-day RS application (*Figure 1H*; 3, 7, or 10 days: p < 0.001), although the calculation of percentage increase in c-Fos-positive cells over CTR group suggests that longer RS induced c-Fos expression in more DCN neurons (*Figure 1—figure supplement 1E*; p = 0.003). Thus, the DCN neurons in the DN were already activated at the early stage of chronic stress, suggesting that the DCN neuron activity is involved in stressful situations.

The above-mentioned results raised the possibility that an increase in activity of DCN neurons in the DN may be relevant to the development of depression-like behaviors. To test this possibility, we aimed to reduce the activity of DCN neurons mainly in the DN during RS application by chemogenetically activating PCs, which provide inhibitory inputs into DCN neurons. Previous studies reported that axons projecting from crus I, crus II, and the simplex lobe are rich in the DN (*Edge et al., 2003*; *Tolbert and Bantli, 1978*; *Voogd et al., 2012*). Among these cortical regions, crus I is well recognized for its involvement in higher brain functions, such as emotion and sociability (*Ferrari et al., 2018*; *Kelly et al., 2020*; *Tsai et al., 2018*; *Turner et al., 2007*). Even though the topographic organization of stress-associated responses in the cerebellum is complex, changes in functional connectivity were observed in crus I of patients with depression (*Alalade et al., 2011*; *Depping et al., 2018*; *Guo et al., 2013*; *Liu et al., 2012*). Hence, we evaluated the effects of the activation of crus I PCs during RS application. To enhance PC activity, hM3Dq (Gq), which is an excitatory designer receptor exclusively activated by designer drugs (DREADD), was expressed in crus I PCs by injecting AAV-sSyn-FLEX-hM3Dq(Gq)–2AGFP into crus I of transgenic mice expressing Cre recombinase exclusively in PCs (PCP2-Cre) (*Figure 2A*). The increase in activity of the Gq-expressing PCs by clozapine-N-oxide (CNO) was first confirmed in vitro. Cell-attached recordings from Gq-expressing PCs in

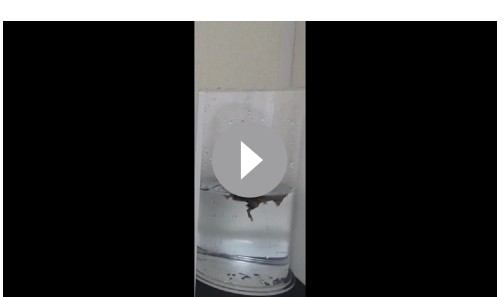

**Video 4.** FST of the RS group.
https://elifesciences.org/articles/72981/figures#video4

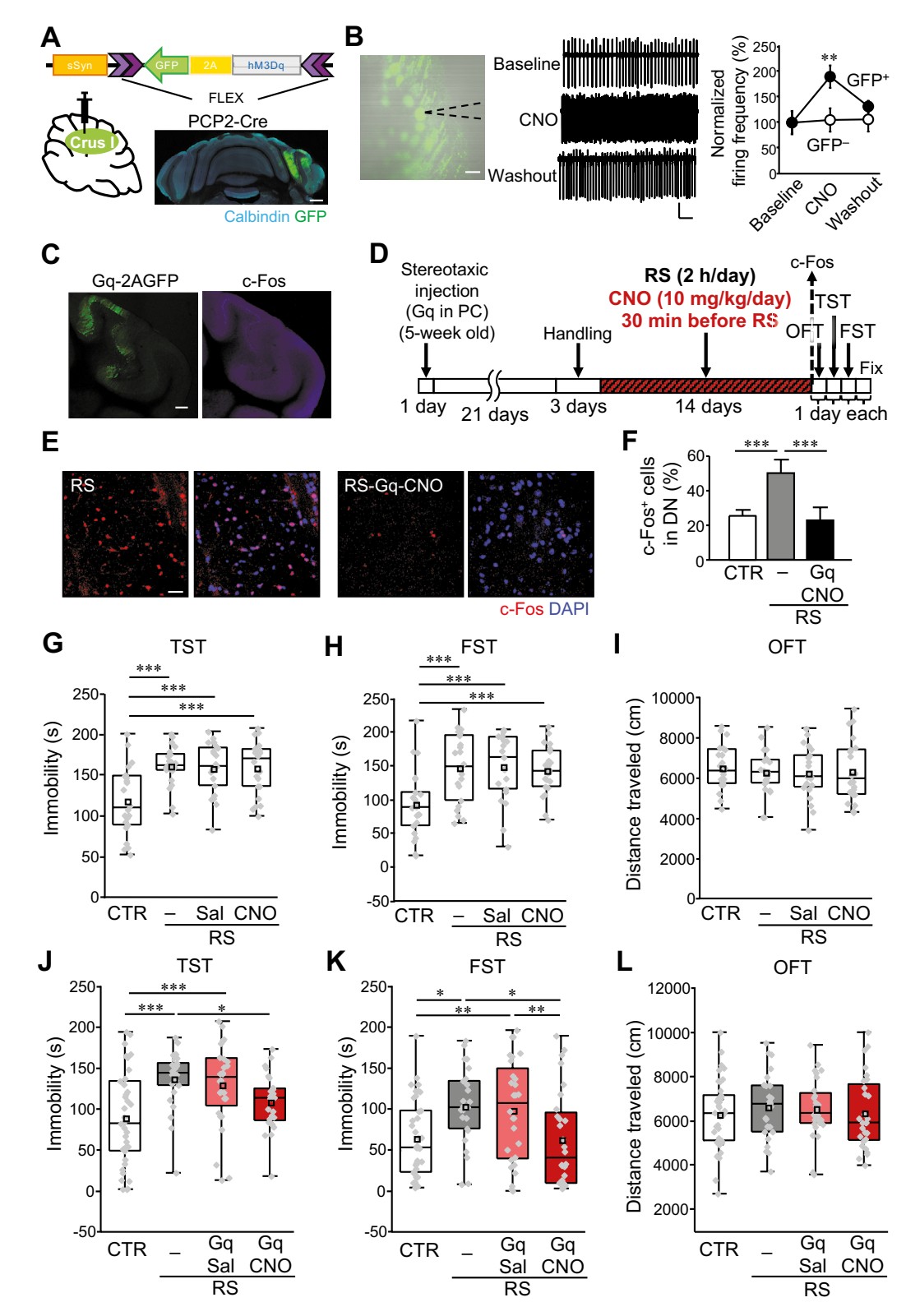

**Figure 2.** Manipulation of crus I PC activity improves depression-like behaviors. (**A**) Top: a schematic of AAV-sSyn-FLEX-hM3Dq-2AGFP injection into crus I of PCP2-Cre mice. Bottom: confocal image of a whole cerebellar slice showing the expression of Gq-2AGFP. Scale bar: 500 μm. (**B**) Left: representative image showing Gq-2AGFP-positive PCs in crus I and a recording pipette (dotted line). Scale bar: 30 μm. Middle: representative traces of PC firing recorded by the loose cell-attached patch-clamp technique, before (top, baseline), during (middle, CNO), and after (bottom, washout)

*Figure 2 continued on next page*

*Figure 2 continued*

CNO treatment. Calibration: 50 pA, 200ms. Right: firing frequency at baseline, during CNO treatment, and at washout normalized by the average value of the baseline in GFP-positive (GFP⁺, closed circles) and GFP-negative (GFP⁻, open circles) PCs (n = 5 GFP⁺ PCs from 4 mice, n = 8 GFP⁻ PCs from 3 mice). (**C**) Enhanced c-Fos expression in crus I PCs (right), upon CNO administration to a mouse expressing Gq-2AGFP in crus I PCs (left). Scale bar: 200 µm. (**D**) Schematic diagram of the experimental time course. RS application (2 hours/day, for 2 weeks) started 3 weeks after stereotaxic AAV injection, followed by 1 day each for the OFT, TST, and FST before fixation, or by fixation for c-Fos staining. CNO (10 mg/kg/day) was intraperitoneally administered 30 min before the RS. (**E and F**) Representative 3D projection images (**E**) and a summary graph (**F**) of c-Fos expression in the DN. Enhanced c-Fos expression in the RS group (left in E, gray bar in F) was reduced by chemogenetic excitation of crus I PCs using Gq expression and CNO administration (right in **E**, black bar in **F**, n = 5 images obtained from 4 mice). Scale bar in E: 20 µm. For direct comparison, results of the DN used in *Figure 1E and G* are shown in (**E**) (RS) and (**F**), respectively. Data are shown as the mean ± SEM. (**G–I**) Assessments by the TST (**G**), the FST (**H**), and the OFT (**I**), to analyze the effects of CNO administration itself on RS-dependent depression-like behaviors in C57BL/6 mice without stereotaxic AAV injection. For a direct comparison, data for the CTR and the RS groups, which were obtained from concurrently performed experiments and shown in *Figure 1B–D*, are reused in this figure. ***p < 0.001, one-way ANOVA followed by Fisher's LSD post hoc test (n = 21 and 22 mice for the RS-Sal and RS-CNO groups, respectively). (**J–L**) Immobility time in the TST (**J**) and FST (**K**), and total distance moved in the OFT (**L**), performed to analyze the effects of chemogenetic excitation of crus I PC activity in PCP2-Cre;ICR mice. *p < 0.05, **p < 0.01, ***p < 0.001, one-way ANOVA followed by Fisher's LSD post hoc test (n = 38, 25, 28, and 28 mice for the CTR, RS, RS-Gq-PC-Sal, and RS-Gq-PC-CNO groups, respectively). Data are presented as boxplots, as described in the legend to *Figure 1*. Exact p values and the statistical tests used are available in *Figure 2—source data 1*. Effect sizes for behavioral data are available in *Figure 2—source data 2* and n numbers for behavioral tests are available in *Figure 2—source data 3*.

The online version of this article includes the following source data and figure supplement(s) for figure 2:

**Source data 1.** p Values and statistical tests related to *Figure 2*.

**Source data 2.** Effect sizes for behavior results related to *Figure 2*.

**Source data 3.** Numbers of mice for each sex used in each behavioral test related to *Figure 2*.

**Figure supplement 1.** Similar effects of unilateral and bilateral induction of crus I PC activity on the development of depression-like behaviors.

**Figure supplement 2.** Changes in body weights of mice used for *Figure 2*.

cerebellar slices showed increased spontaneous firing after CNO application (10 µM) compared with the baseline, whereas the PCs in the uninfected cerebellum demonstrated no effects of CNO application (*Figure 2B*; Gq-PCs: $t_{(8)}$ = 3.7, p = 0.003; uninfected PCs: $t_{(14)}$ = 0.19, p = 0.85). The increase in activity was also confirmed in vivo by immunohistochemical analysis of c-Fos, which showed that intraperitoneal administration of CNO into mice expressing Gq in PCs resulted in an increased c-Fos level in Gq-expressing PCs in crus I (*Figure 2C*). In addition, when we used this chemogenetic tool for enhancing crus I PC activity by the daily injection of CNO for the 2 weeks prior to RS application (*Figure 2D*), c-Fos expression in the DN was reduced compared with when only RS was applied (*Figure 2E*). Quantified data showed that c-Fos-positive cells in the DN after chronically enhancing the activity of crus I PCs during RS were significantly decreased and the level was equivalent to the control (*Figure 2F*; RS – Gq-CNO: $t_{(11)}$ = –4.25, p < 0.001; CTR – Gq-CNO: $t_{(11)}$ = –0.43, p = 0.67). Although this way of crus I PC manipulation might induce plasticity in other subregions of the DCN, as crus I PCs are also known to project to the IPN (*Sugihara, 2018*), this manipulation certainly includes the activity control of the DN, leading to the restoration of activity in the DN that was increased by the RS. Thus, our chemogenetic manipulation of PCs appeared to be an experimental system applicable to test the involvement of the cerebellum in the development of stress-induced behavioral changes.

In the behavioral experiments of this study, in which we analyzed the effects of chemogenetic manipulation on RS-dependent depression-like behaviors, AAVs expressing chemogenetic molecules were stereotaxically injected 3 weeks before the RS procedures, and CNO or saline was intraperitoneally administered 30 min before the RS application every day during the 2-week RS period (*Figure 2D*), unless stated otherwise. To enable the comparison within a series of behavioral experiments even in the presence of mild variability between different series, we included the CTR and the RS groups in all series of concurrently performed behavioral experiments, and compared groups of saline or CNO administration to the CTR and the RS groups. To test off-target effects of CNO on the RS-dependent depression-like behaviors, saline (RS-Sal) or CNO (RS-CNO) was administered to the mice that were subjected to RS without the expression of chemogenetic molecules. The RS-dependent behavioral changes were not affected by either administration of saline or CNO, as seen by the significantly longer immobile times than the CTR group, but the similar immobile times to the RS group, in both the TST and FST (*Figure 2G and H*; CTR – RS-Sal, TST: $t_{(41)}$ = 3.91, p < 0.001, FST: $t_{(41)}$ = 3.88, p < 0.001; CTR – RS-CNO, TST: $t_{(42)}$ = 4.01, p < 0.001, FST: $t_{(42)}$ = 3.52, p < 0.001; RS – RS-Sal, TST: $t_{(41)}$ = –0.29, p

= 0.77, FST: $t_{(41)}$ = 0.12, p = 0.9; RS – RS-CNO, TST: $t_{(42)}$ = –0.24, p = 0.81, FST: $t_{(42)}$ = –0.29, p = 0.78). The travel distance measured by the OFT was also unaffected by the saline or CNO administration (*Figure 2I*; CTR – RS-Sal, $t_{(41)}$ = –0.68, p = 0.5; CTR – RS-CNO, $t_{(42)}$ = –0.46, p = 0.65; RS – RS-Sal, $t_{(41)}$ = –0.11, p = 0.91; RS – RS-CNO, $t_{(42)}$ = 0.12, p = 0.91). These results confirm that the intraperitoneal administration of saline or CNO itself has no effects on the RS-dependent depression-like behaviors. In contrast, CNO administration to the PCP2-Cre mice expressing Gq specifically in the PCs of crus I, referred to as the RS-Gq-PC-CNO group, but not saline administration (RS-Gq-PC-Sal group), shortened the immobile time in both the TST and FST (*Figure 2J and K*). As unilateral and bilateral excitation of crus I PCs resulted in similar trends in all the TST, FST, and OFT, as shown in *Figure 2—figure supplement 1A–1F*, combined results are presented in the main figures (*Figure 2J–L*). Even though there was considerable individual variability in all groups, the average immobile time in the RS-Gq-PC-CNO group was significantly shorter than that of the RS group, but was indistinguishable to the CTR group (*Figure 2J and K*; RS – RS-Gq-PC-CNO, TST: $t_{(51)}$ = –2.18, p = 0.03, FST: $t_{(51)}$ = –2.75, p = 0.007; CTR – RS-Gq-PC-CNO, TST: $t_{(64)}$ = 1.66, p = 0.1, FST: $t_{(64)}$ = –0.12, p = 0.91). The immobile time in RS-Gq-PC-Sal group was conversely indistinguishable to that of the RS group, but was significantly longer than the CTR group (*Figure 2J and K*; RS – RS-Gq-PC-Sal, TST: $t_{(51)}$ = –0.57, p = 0.57, FST: $t_{(51)}$ = –0.33, p = 0.74; CTR – RS-Gq-PC-Sal, TST: $t_{(64)}$ = 3.45, p < 0.001, FST: $t_{(64)}$ = 2.55, p = 0.01). The OFT showed unchanged distance movement in all four groups (*Figure 2L*; $F_{(3,115)}$ = 0.33, p = 0.33), ruling out the possibility that the different immobile times arose from altered general locomotor activity. These results in behavior, together with the c-Fos data, suggest that the increase in activity of DCN neurons driven by the chronic mild stress promotes the generation of depression-like behaviors. Consequently, the excitatory manipulation of crus I PC activity during RS application suppressed the generation of depression-like behaviors by inhibiting the activity of DCN neurons in the DN. We hence conclude that cerebellar neuronal activity indeed contributes to the stress-dependent development of depression-like behaviors.

## DN neurons anatomically project to the VTA

Whereas our results showed the involvement of the cerebellum in the development of depression-like behaviors, the precise circuit involving the cerebellum remains unclear. As previous studies reported axonal inputs of the DCN in the VTA (*Parker et al., 2014*; *Beier et al., 2015*; *Carta et al., 2019*) which is a region highly associated with stress responses (*Fox and Lobo, 2019*), we further confirmed the network connections from the DCN to the VTA. DCN neurons in each subregion were labeled by iontophoretic delivery (*Oh et al., 2014*) of an adeno-associated virus (AAV) expressing GFP (AAV-sSyn-GFP) into the right side of each subregion, which resulted in GFP expression in restricted regions of the DCN (see the cerebellum in *Figure 3A–C*). Upon injection into the FN, GFP-positive axonal projections were not observed in the VTA (*Figure 3A*, *Figure 3—figure supplement 1A*). Injection to the IPN clearly displayed GFP-positive axons in the red nucleus (RN), which is a well-known projection area from the cerebellum (*Figure 3B*, *Figure 3—figure supplement 1B*). However, there were only weak GFP-positive axonal projections from the IPN in the contralateral side of the dorsolateral anterior VTA and slightly stronger GFP signals in the dorsolateral posterior VTA (*Figure 3B*, *Figure 3—figure supplement 1B*). While the injection into the DN displayed strong GFP-positive axons in the RN, it also resulted in GFP signals mainly in the contralateral side of both the anterior and posterior VTA, and the signals were most abundant compared with the injection into the other two nuclei (*Figure 3C*, *Figure 3—figure supplement 1C*). Particularly, DCN neurons in the DN projected mainly to the dorsolateral posterior VTA among all VTA regions. Thus, consistent with previous studies (*Parker et al., 2014*; *Carta et al., 2019*), our results showed the projection from the DN and the IPN, but not the FN, to the VTA, and also suggested that the majority of DCN projections to the VTA are from the DN. This was further confirmed by two other experiments using a retrograde transducing AAV, rAAV2-retro (*Tervo et al., 2016*). When rAAV2-retro expressing GFP (rAAV2-retro-CAG-sfGFP) was iontophoretically delivered into the posterior VTA (*Figure 3D*), faint but clear GFP signals were detected in many DCN neurons of the DN, but in fewer neurons of the IPN (*Figure 3E*). In another set of experiments, rAAV2-retro expressing Cre recombinase (rAAV2-retro-CAG-iCre) was injected into the posterior VTA, and AAV expressing GFP with a Cre-dependent genetic switch (AAV-sSyn-FLEX-GFP) was injected over a broad area of the contralateral DCN, including the DN and the IPN (*Figure 3F*). This combination of AAV injection resulted in strong GFP signals that were observed

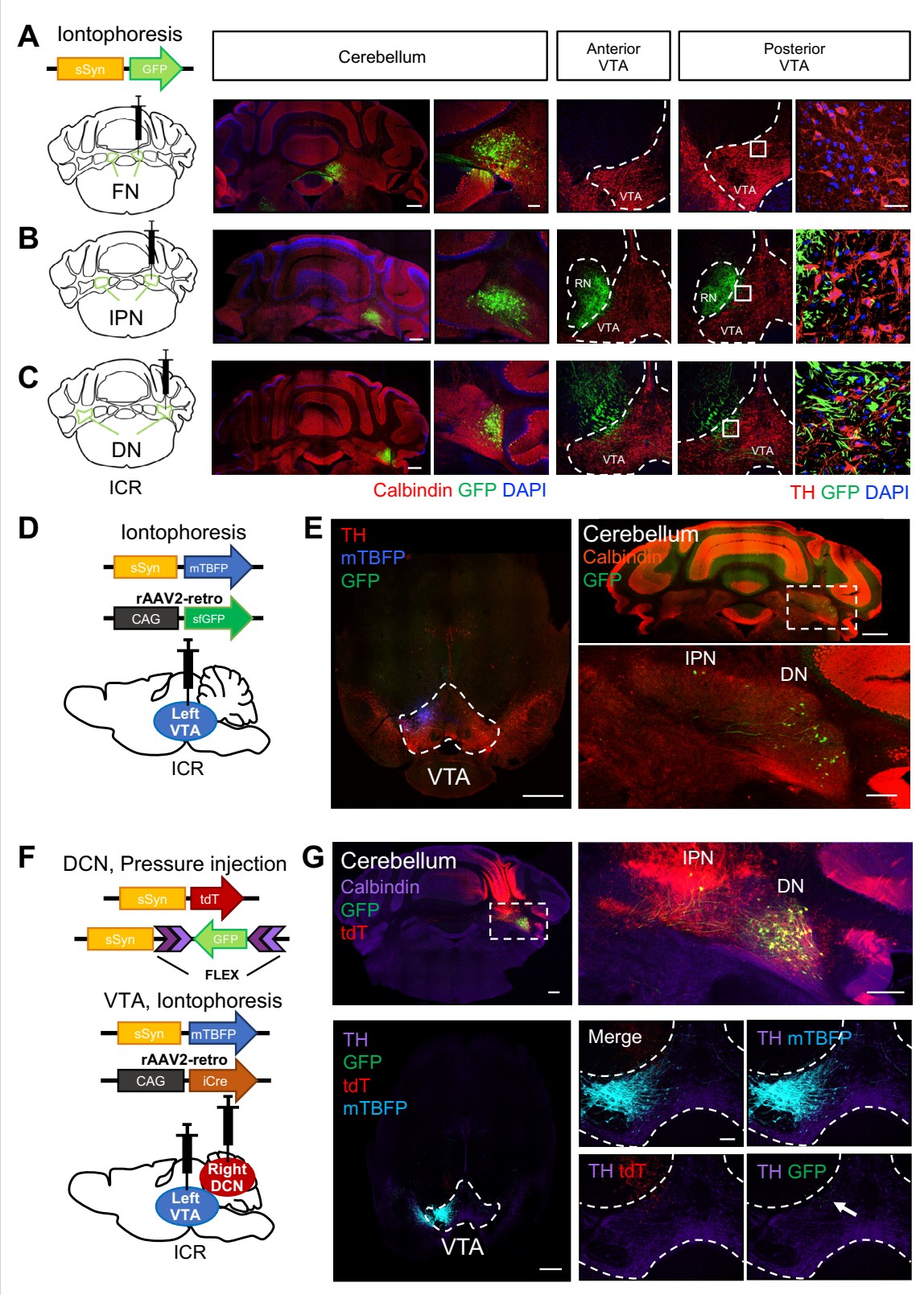

**Figure 3.** DCN neurons in the DN structurally project to the VTA. (**A–C**) Left: schematic drawing of AAV-sSyn-GFP injection into the right side of the FN (**A**), the IPN (**B**), and the DN (**C**). Note that iontophoretic injection was used, to achieve confined injection of AAV in the individual nuclei. Right: confocal images showing the resulting GFP expression in cerebellar slices stained with a calbindin antibody and VTA slices stained with a tyrosine hydroxylase (TH) antibody. The rightmost panels are 3D projection views of the boxed areas. Scale bars: 500 μm (left), 50 μm (middle), and 20 μm (right).

*Figure 3 continued on next page*

*Figure 3 continued*

Similar image results are obtained from 7, 8, and 4 mice for fastigial, interposed and dentate nuclei respectively. (**D**) Schematic drawing of iontophoretic injection of rAAV2-retro-CAG-sfGFP and AAV-sSyn-mTagBFP (mTBFP) into the left side of the VTA. (**E**) Confocal images of a VTA-containing slice stained with a TH antibody (left) and a cerebellar slice stained with a calbindin antibody (right). Expression of GFP and mTBFP resulting from the injection shown in (**D**). The area within the white dotted rectangle shown in the top right panel is magnified in the bottom right panel. Scale bars: 500 μm (left and top right), and 200 μm (bottom right). For retrograde single injection, similar image results are obtained from 3 mice. (**F**) Schematic diagram of combined AAV injection. AAV-sSyn-tdT and AAV-sSyn-FLEX-GFP were injected into the right side of the DCN by pressure injection, whereas AAV-sSyn-mTBFP and rAAV2-retro-CAG-iCre were injected into the left side of the VTA by iontophoresis. (**G**) Representative confocal images of a cerebellar slice stained with a calbindin antibody (top) and a VTA-containing midbrain slice stained with a TH antibody (bottom). Expression of GFP, tdT, and mTBFP resulting from the injection shown in (**F**). The right panels show magnified images of the DCN (the white dotted rectangle) and VTA, including GFP-positive axons of DCN neurons (arrow). Scale bars: 500 μm (left), 250 μm (top right), and 100 μm (bottom right). For double injections, similar image results are obtained from 10 mice. White dotted rectangles shown in (**E**) and (**G**) indicate location of magnified images for DCN areas.

The online version of this article includes the following figure supplement(s) for figure 3:

**Figure supplement 1.** Anatomical connectivity from each subregion of the DCN to the VTA.

in many DCN neurons of the DN, but in only a few neurons of the IPN (*Figure 3G*, top). The small numbers of GFP-positive neurons in the IPN did not appear to be due to the less efficient delivery of AAV into the IPN, because tdTomato (tdT) signals derived from the simultaneous injection of AAV-sSyn-tdT was observed similarly in both the DN and IPN (*Figure 3G*). Through the process of double injection (*Figure 3F*), we were able to specifically label DCN neurons that project to the VTA. These results of AAV-based tracing confirm that the VTA receives anatomical connectivity of DCN neurons mainly from the DN among all three DCN subregions, even though the VTA-projecting DCN neurons seemed to be only a part of all DCN neurons in the DN, as seen in the images of the cerebellum in *Figure 3E and G*. In the subsequent experiments of this study, in which the VTA-projecting DCN neurons express exogenous molecules by AAV injection, we mainly targeted the DN among all DCN subregions, although we understand that DCN neurons in the IPN may also be involved.

## DCN axons form functional synaptic connections to the VTA neurons

We next aimed to identify whether the DCN neurons in the DN form functional synaptic connections with VTA neurons, and performed whole-cell patch clamp recording of VTA neurons after a single injection of AAV expressing channelrhodopsin (ChR2) into the DN (*Figure 4A*). VTA neurons, mainly located in the dorsolateral part where ChR2-positive axons coming from the DCN are present, were voltage clamped at −70 mV, and excitatory postsynaptic currents (EPSCs) were recorded by 5 trains of blue light stimuli (10 or 50 Hz) applied onto ChR2-expressing DCN axons in midbrain slices (*Figure 4B*). This photostimulation triggered EPSCs in VTA neurons (*Figure 4C and D*, *Figure 4—figure supplement 1A*). Although the photostimulation-triggered EPSCs were detected in only some of the VTA neurons (13 out of 54 neurons, *Figure 4E*), the responses were clearly and repeatedly observed upon photostimulation (*Figure 4C and D*). To test whether the photostimulation-triggered EPSCs are evoked by monosynaptic connections, we used inhibitors of voltage-gated sodium channels, tetrodotoxin (TTX), and voltage-gated potassium channels, 4-aminopyridine (4-AP), as was done previously (*Carta et al., 2019*; *Yan et al., 2019*). The application of TTX inhibited the photostimulation-triggered EPSCs, yet subsequent addition of 4-AP surmounted the TTX-dependent inhibition of EPSCs (*Figure 4F*; baseline – TTX: $t_{(8)} = 3.69$, p = 0.006; TTX – TTX + 4-AP: $t_{(8)} = 2.63$, p = 0.03). Because 4-AP is considered to enhance depolarization of photostimulated ChR2-expressing axons and consequently induce neurotransmitter release even in the presence of TTX, the results indicate that the connections from the DCN to the VTA are monosynaptic.

It is well known that VTA neurons include different types of neurons. One way to distinguish and identify DA neurons in the VTA is to measure the hyperpolarization-activated current ($I_h$) with negative voltage steps (e.g. −110 mV) (*Borgland et al., 2004*; *Faleiro et al., 2004*; *Bellone and Lüscher, 2006*; *Hommel et al., 2006*; *Argilli et al., 2008*; *Stuber et al., 2008*; *Zweifel et al., 2008*), although the presence of $I_h$ current does not unequivocally identify DA neurons (*Margolis et al., 2006*; *Zhang et al., 2010*; *Lammel et al., 2011*). To validate the method of determining VTA cell types by measuring $I_h$, we recorded $I_h$ currents in GAD2-IRES-Cre;Ai6 mice, which express ZsGreen in γ-aminobutyric acid (GABA)-ergic neurons, and quantified the amplitude (*Figure 4—figure supplement 1B*). Indeed, $I_h$ currents were not observed in most of ZsGreen-positive neurons whereas ZsGreen-negative neurons

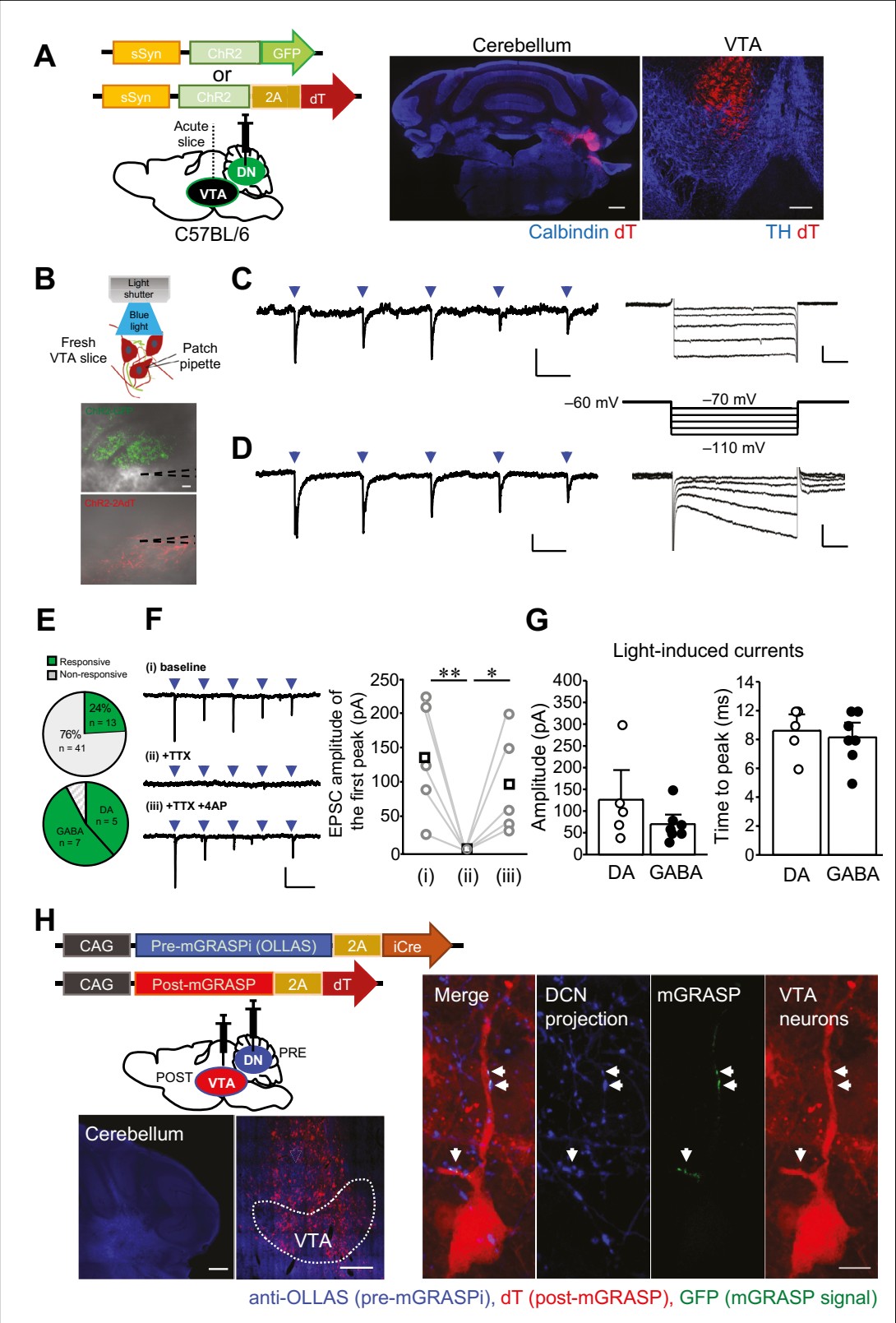

**Figure 4.** The DCN neurons in the DN make functional synaptic connections to the VTA. (**A**) Left: schematic drawing of the injection of AAV-sSyn-ChR2-GFP or AAV-sSyn-ChR2-2AdT into the DN, for whole-cell patch clamp recording from VTA neurons. Right: confocal images of ChR2-2AdT expression in a calbindin-stained cerebellar slice (left) and a TH-stained VTA slice (right). Scale bars: 500 µm (left), and 200 µm (right). (**B**) Schematic diagram of whole-cell patch clamp recording to detect light-evoked EPSCs in VTA neurons, and images of fresh midbrain slices containing the VTA with ChR2-

*Figure 4 continued on next page*

*Figure 4 continued*

GFP- or ChR2-2AdT-positive axons projecting from the DN. Dotted lines in the images indicate the recording pipettes. Scale bar: 20 μm. (**C and D**) Representative traces of EPSCs evoked by 10 Hz photostimulation at a holding potential of −70 mV. To measure $I_h$ currents, voltage steps varying from −70 mV to −110 mV were applied from a holding potential of −60 mV. Calibration: 20 pA, 50ms for light-evoked EPSC; 50 pA, 100ms for $I_h$ currents. (**E**) Pie charts showing the proportion of VTA neurons responding to photostimulation among all VTA neurons recorded (top, 54 cells obtained from 9 mice), and the ratio of DA and GABA neurons among VTA neurons demonstrating light-evoked EPSCs (bottom). (**F**) Representative traces (left) and peak amplitude (right) of light-evoked EPSCs before (baseline) and after extracellular application of TTX (+ TTX) followed by further addition of 4-AP (+ TTX + 4 AP). *p < 0.05, **p < 0.01, One-way repeated measures ANOVA followed by Fisher's least significant difference (LSD) post hoc test. Calibration: 100 pA, 100ms. Averaged (squares) and individual (circles) EPSC amplitudes are shown on the right (n = 5 from 3 mice). (**G**) Peak amplitudes (left) and time to peak (right) of light-evoked EPSCs recorded from DA and GABA neurons. Data are shown as the mean ± SEM. Exact *p* values and the statistical tests used are available in *Figure 4—source data 1*. (**H**) Detection of synaptic connections from the DN to the VTA using mGRASPi. Left: schematic diagram showing injections of AAVs expressing pre-mGRASPi(OLLAS)–2AiCre and post-mGRASPi-2AdT into the DN and the VTA, respectively (top), and subsequent observation of immunostained OLLAS signals in the DN and dT signals in the VTA (bottom). Scale bar: 200 μm. Right: magnified images of a VTA neuron surrounded by DCN axons expressing the pre-mGRASP component. Note that mGRASP signals (arrows) were clearly observed where button-like structures of the DCN axons intersect the VTA neuron dendrites. Scale bar: 10 μm.

The online version of this article includes the following source data and figure supplement(s) for figure 4:

**Source data 1.** p Values and statistical tests related to *Figure 4*.

**Figure supplement 1.** Functional synaptic connections of the DCN-VTA circuit.

showed clear $I_h$ currents (*Figure 4—figure supplement 1C*; p = 0.0002). While we are aware that this is not the definitive approach to identify VTA cell types, as seen in our result showing $I_h$ currents in a ZsGreen-positive neuron (*Figure 4—figure supplement 1C*), we distinguished putative DA and GABA neurons by testing the $I_h$ current in this study. Photostimulation-evoked EPSCs were recorded in both putative GABA (*Figure 4C*) and DA neurons (*Figure 4D*). Except for one neuron that was unidentified due to the membrane seal being lost, 5 and 7 neurons exhibiting EPSCs were identified as putative DA and GABA neurons, respectively (*Figure 4E*), indicating the nonspecific projection of DCN neurons in the DN to both types of VTA neurons. This is consistent with previous studies showing connections from the DCN to the VTA (*Beier et al., 2015*; *Carta et al., 2019*). The peak amplitude and time course of photostimulation-evoked EPSCs were equivalent between putative DA and GABA neurons (*Figure 4G*; peak amplitude: p = 0.21; time course: p = 0.66). Furthermore, structural synaptic connections of the DN-VTA circuit were verified through using the improved version of mammalian GFP reconstitution across synaptic partners (mGRASPi) technique, which enabled us to detect synaptic contacts using light microscopy (*Kim et al., 2011*). AAV vectors expressing presynaptic-mGRASP (AAV-CAG-pre-mGRASPi(OLLAS)–2AiCre) and postsynaptic-mGRASP (AAV-CAG-post-mGRASPi-2AdT) components carrying either of the nonfluorescent split-GFP fragments were injected into the DN and the VTA, respectively. As a result, reconstituted mGRASP signals were detected between the button-like structures of anti-OLLAS-positive presynaptic DCN neuronal axons and a dT-positive postsynaptic VTA neuron (*Figure 4H*), confirming DN-VTA synaptic connections. These results of electrophysiological and mGRASP analyses demonstrated that DCN neurons in the DN make functional synaptic connections to putative DA and GABA neurons in the VTA.

## Crus I PCs form functional synaptic connections to VTA-projecting DCN neurons in the DN

Even though the DCN is the final output structure of the cerebellum and projects to other brain regions, it also projects back to the cerebellar cortex. It is known that both corticonuclear and nucleocortical pathways between the cerebellar cortex and DCN are similarly organized into sagittal zones, and DCN neurons in the DN usually have reciprocal connections with the lateral lobes (*Houck and Person, 2014*). To clarify whether VTA-projecting DCN neurons, located mainly in the DN, also have collateral projections to the lateral lobes of the cerebellar cortex, we analyzed collateral axons of the VTA-projecting DCN neurons that were specifically labeled by the double injection method described in *Figure 3F*. Whereas GFP-positive DCN neurons were in the DN and their axons were in the VTA (*Figure 5A and B*), screening of the entire cerebellar cortex demonstrated that their collateral axons with rosette shaped mossy fiber terminals (MFTs) were mainly present in crus I and were also present in crus II and the simplex lobe (*Figure 5C*, *Figure 5—figure supplement 1A*). These were specific GFP signals arising from VTA-projecting DCN neurons in spite of the sparse GFP signals in collateral

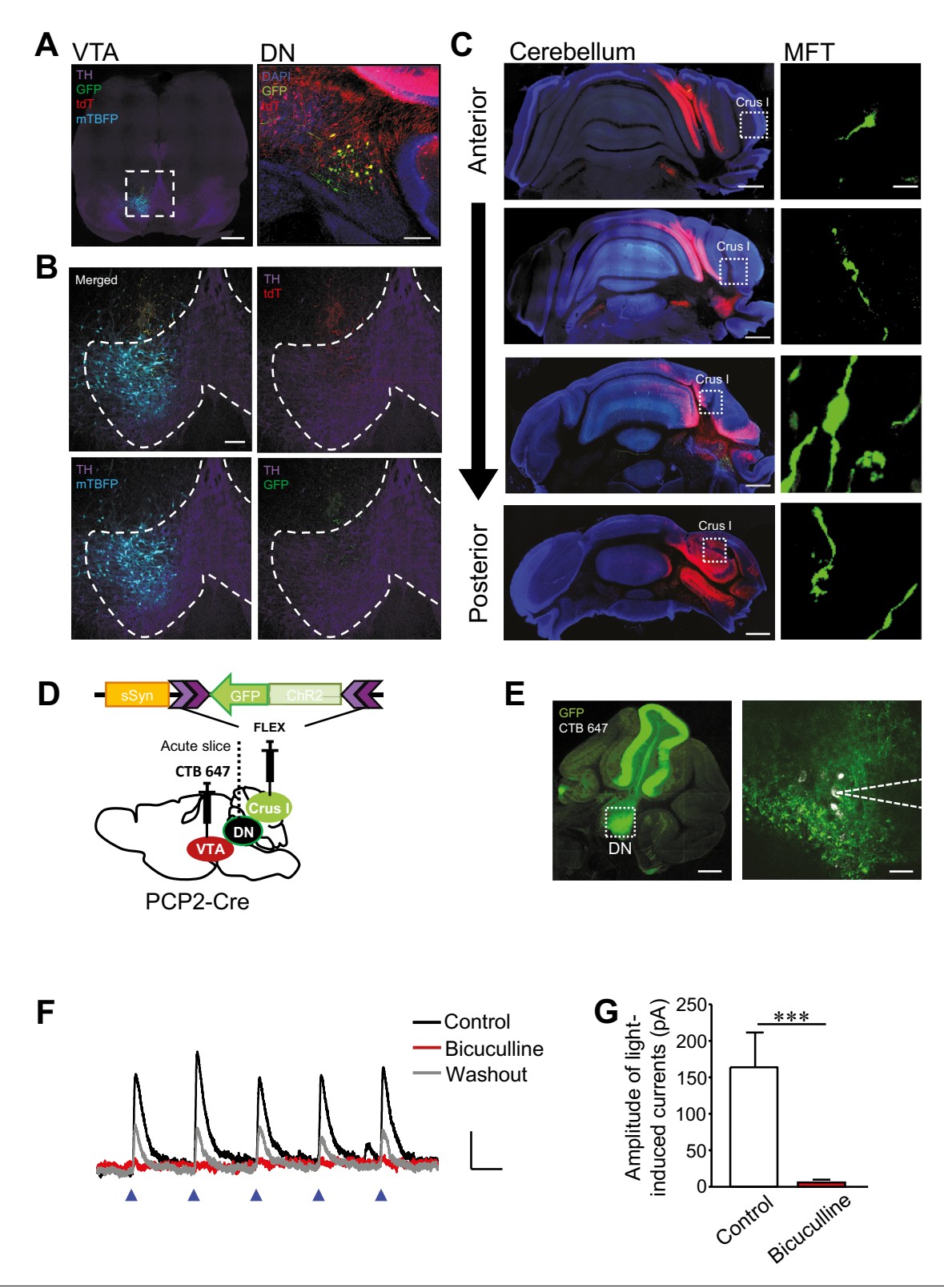

**Figure 5.** Corticonuclear and nucleocortical connections between crus I and VTA-projecting DCN neurons in the DN. (**A and B**) Representative images of a midbrain slice (left) and a cerebellar slice (right) in (**A**). In the left panel, the area within the white dotted square is magnified in (**B**), and dotted lines in (**B**) indicate the boundary of the VTA region. Expression of tdT, mTBFP, and GFP was triggered by the AAV injection shown in *Figure 3F*. Scale bars: 500 μm (left in **A**), 100 μm (right in **A** and **B**). (**C**) Screening of collateral axons of VTA-projecting DCN neurons in the cerebellar cortex. Cerebellar slices

*Figure 5 continued on next page*

Figure 5 continued

were obtained from the same mouse used for the images in (**A**) and (**B**). A MFT within the white dotted square of each panel on the left is magnified on the right. Scale bars: 500 μm (left), 10 μm (right). (**D**) Diagram of the injection of AAV-sSyn-FLEX-ChR2-GFP into crus I and CTB 647 into the VTA of PCP2-Cre mice. Fresh cerebellar slices were then prepared to record the synaptic transmission from VTA-projecting DCN neurons in the DN upon the photostimulation of ChR2-positive PC axons. (**E**) Resultant expression of ChR2-GFP in PCs and CTB 647 labeling in the DN of a cerebellar slice in sagittal section. The area within the white dotted square of the left panel is magnified in the right panel. Whole-cell patch-clamp recording (dotted lines on the right) was performed from CTB 647-positive neurons, as seen in the magnified image on the right. Scale bars: 250 μm (left), 40 μm (right). (**F**) Representative traces of IPSCs evoked by 10 Hz photostimulation at a holding potential of −50 mV, before (control), during (bicuculline), and after (washout) bath application of bicuculline. Calibration: 20 pA, 50ms. (**G**) Peak amplitudes of light-evoked IPSCs. ***p < 0.001, Student's *t*-test (n = 9 and 5 for control and bicuculline, respectively). Data are shown as the mean ± SEM. Exact p values and the statistical tests used are available in *Figure 5— source data 1*.

The online version of this article includes the following source data and figure supplement(s) for figure 5:

**Source data 1.** p Values and statistical tests related to *Figure 5*.

**Figure supplement 1.** Nucleocortical connections of the VTA-projecting DCN neurons.

axons, as there were no clear GFP signals detected in collateral axons without injection into the VTA (*Figure 5—figure supplement 1B*). These results indicate that crus I, II, and the simplex lobe receive feedback signals from DCN neurons in the DN, which specifically project to the VTA.

As the results of nucleocortical pathways suggested that the lateral lobes likely send inputs to the VTA-projecting DCN neurons based on the zonal organization, we further investigated whether the VTA-projecting DCN neurons in the DN actually receive inputs from PCs of crus I that was associated with stress-induced behavioral changes (*Figure 2*). For this purpose, we recorded optogenetically evoked inhibitory postsynaptic currents (IPSCs) from VTA-projecting DCN neurons that were labeled by the retrograde tracer Alexa Fluor 647-conjugated cholera toxin B (CTB 647), after expressing ChR2 in crus I PCs (*Figure 5D and E*). IPSCs were detected upon the application of five trains of blue light stimuli to the DN in cerebellar sagittal slices (*Figure 5F*). When the same slices were treated with 10 μM bicuculline, which is a GABA type A receptor antagonist, the optically induced IPSCs were completely abolished (*Figure 5F and G*; p = 0.001), confirming inhibitory synaptic transmission from PCs. These results verified the zonal organization of the corticonuclear and nucleocortical pathways in the VTA-associated cerebellar networks, and also identified that crus I PCs send synaptic inputs to DCN neurons in the DN that in turn project to the VTA.

## Activity of VTA-projecting DCN neurons in the DN regulates the development of depression-like behaviors

The above-mentioned results demonstrated that RS-mediated activity increase in DCN neurons of the DN and depression-like behaviors were reduced by excitation of crus I PCs, which have functional connections to the VTA via the DN. Based on these results, we hypothesized that the neuronal circuit composed of DCN neurons projecting to the VTA is responsible for the cerebellar contribution to the stress-dependent development of depression-like behaviors. To directly test this hypothesis, we suppressed the activity of VTA-projecting DCN neurons in the DN during RS application. The inhibitory DREADD hM4Di (Gi) was expressed in VTA-projecting DCN neurons in the DN by combining the injection of rAAV2-retro-CAG-iCre into the dorsolateral posterior VTA of wild-type mice with bilateral injection of AAV-sSyn-FLEX-hM4Di(Gi)–2AGFP into the DN (*Figure 6A*). As in the case of chemogenetic activation of PCs, CNO or saline was intraperitoneally administered 30 min before the RS every day for the 2-week RS period. Whereas saline administration (RS-Gi-DN-Sal group) did not affect the immobility time prolonged by the RS in the TST and FST, CNO administration (RS-Gi-DN-CNO group) shortened the immobility time. As a result, the averaged immobility time in the RS-Gi-DN-CNO group was significantly shorter than that of the RS-Gi-DN-Sal group, and was similar to the CTR group in both the TST and FST (*Figure 6B and C*; RS-Gi-DN-Sal – RS-Gi-DN-CNO, TST: $t_{(38)}$ = −2.23, p = 0.03, FST: $t_{(38)}$ = −3.2, p = 0.002; CTR – RS-Gi-DN-CNO, TST: $t_{(52)}$ = 0.03, p = 0.98, FST: $t_{(52)}$ = 0.94, p = 0.35). In the OFT, there were no differences in the distance traveled among all groups, demonstrating intact locomotor activity of the mice (*Figure 6D*; $F_{(3,112)}$ = 0.71, p = 0.55).

Although the TST and FST have been widely used to assess the depression-like behavior (*Kim and Han, 2006*; *Park et al., 2010*; *Kim and Leem, 2014*; *Zou et al., 2015*; *Wang et al., 2018*; *Planchez et al., 2019*; *Son et al., 2019*; *Gadotti et al., 2019*; *Chevalier et al., 2020*), one may wonder whether

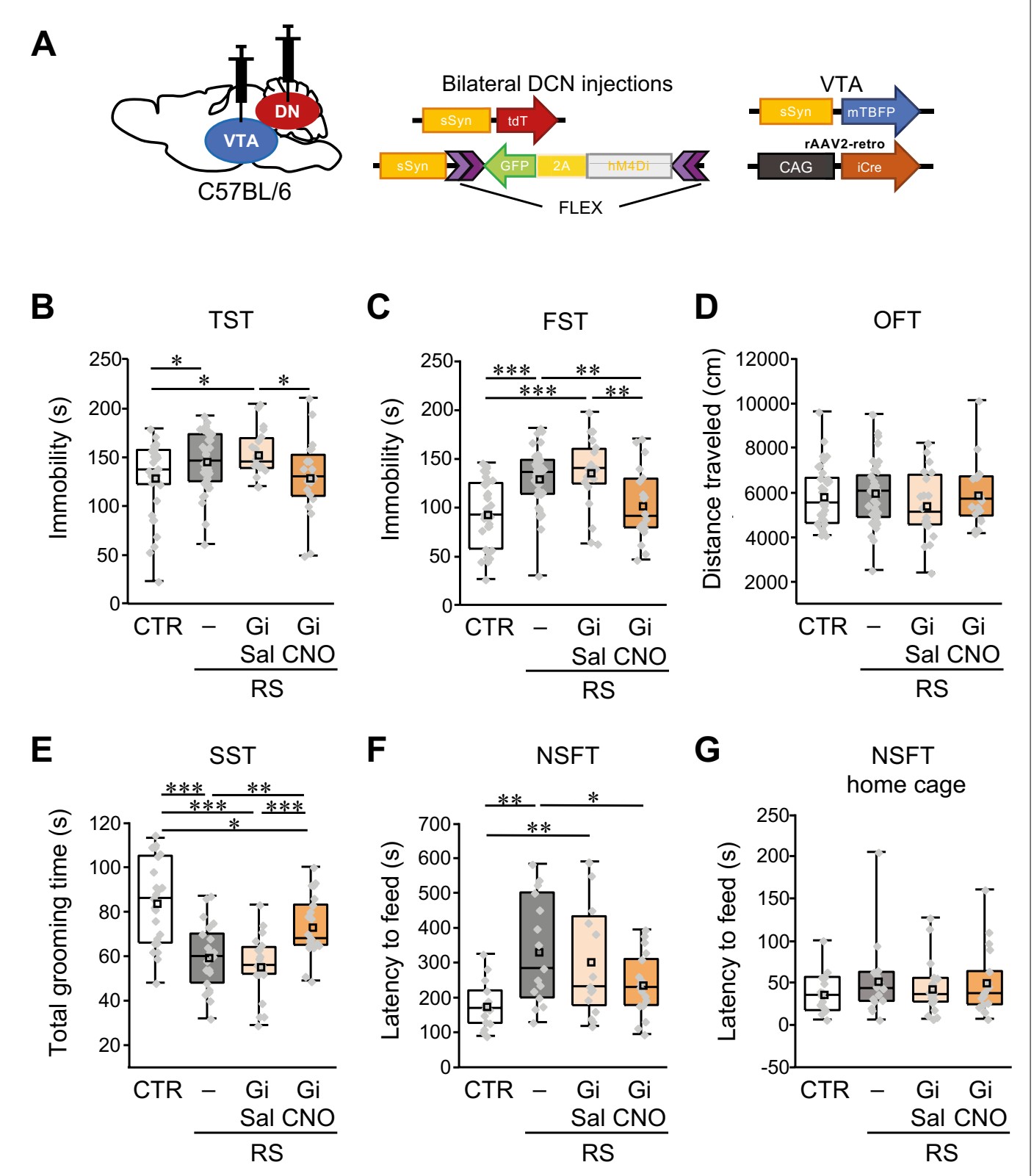

**Figure 6.** Specific inhibition of VTA-projecting DCN neurons in the DN ameliorates depression-like behaviors. (**A**) Diagram of combined AAV injection. Bilateral injections of AAV-sSyn-tdT and AAV-sSyn-FLEX-hM4Di(Gi)–2AGFP was carried out into the DCN and a mixture of AAV-sSyn-mTBFP and retroAAV-CAG-Cre was injected into the VTA. (**B–G**) Immobility time in the TST (**B**) and the FST (**C**), total distance moved in the OFT (**D**), grooming time in the SST (**E**), and latency to feed in the NSFT (**F**) and in the home cage (**G**), to see the effects of the chronic inhibition of VTA-projecting DCN

*Figure 6 continued on next page*

*Figure 6 continued*

neurons using Gi during RS application in C57BL/6 mice. *p < 0.05, **p < 0.01, ***p < 0.001, one-way ANOVA followed by the Fisher's LSD post hoc test (numbers of mice used for the CTR, RS, RS-Gi-DN-Sal, and RS-Gi-DN-CNO groups; n = 34, 42, 20, and 20 mice in the TST, FST, and OFT; n = 23, 21, 20, and 21 mice in the SST; n = 15, 15, 16, and 17 in the NSFT). Data are presented as boxplots, as described in the legend to *Figure 1*. Exact *p* values and the statistical tests used are available in *Figure 6—source data 1*. Effect sizes for behavioral data are available in *Figure 6—source data 2* and n numbers for behavioral tests are available in *Figure 6—source data 3*.

The online version of this article includes the following source data and figure supplement(s) for figure 6:

**Source data 1.** p Values and statistical tests related to *Figure 6*.

**Source data 2.** Effect sizes for behavior results related to *Figure 6*.

**Source data 3.** Numbers of mice for each sex used in each behavioral test related to *Figure 6*.

**Figure supplement 1.** Changes in body weights of mice used for *Figure 6*.

these results truly represent depression-like behaviors, considering that differences in averaged values are significant but small despite large individual variability, and that both tests rely on immobility to evaluate the same dimension of depression-like behaviors, that is despair or hopelessness. Two further analyses were thus conducted. Firstly, we examined the correlation of individual data to see the inter-relationship in different combinations among three behavioral tests. There was a significant correlation between TST and FST, but not between OFT and TST or FST (*Table 1*; TST – FST: $r = 0.33$, $p < 0.001$; TST – OFT: $r = –0.12$, $p = 0.2$; FST – OFT: $r = 0.05$, $p = 0.06$), indicating that the TST and FST measured share the behavioral phenotypes, likely the depression-like behaviors. These results further confirm that immobility level is uncorrelated to locomotor disability, and present the possibility that individual variability may in part arise from degree of depression-like behaviors. Secondly, we tested the involvement of VTA-projecting DCN neurons in other dimensions of the depression-like behaviors using the sucrose splash test (SST) and the novelty suppressed feeding test (NSFT), which have been used to test self-neglect and hyponeophagia or anhedonia, respectively (*Zou et al., 2015*; *Wang et al., 2018*; *Gadotti et al., 2019*; *Planchez et al., 2019*; *Chevalier et al., 2020*). The 2-week RS application significantly decreased the time spent grooming in the SST (*Figure 6E*; $t_{(42)} = –5.12$, $p < 0.001$), and significantly increased the latency to feed in the NSFT (*Figure 6F*; $t_{(28)} = 3.39$, $p = 0.001$), without altering home cage food consumption (*Figure 6G*; $t_{(28)} = 1.14$, $p = 0.26$). These results confirm depression-like behaviors triggered by our RS application protocol. The RS-Gi-DN-Sal group showed similar depression-like behaviors to the RS group in both the SST and NSFT (SST: $t_{(39)} = –0.84$, $p = 0.4$; NSFT: $t_{(29)} = –0.626$, $p = 0.53$). On the other hand, depression-like behaviors were reduced in the RS-Gi-DN-CNO group: mice in the RS-Gi-DN-CNO group spent significantly longer time in grooming in the SST, and took significantly less time to feed in the NSFT, compared with the RS group (*Figure 6E and F*; SST: $t_{(40)} = 2.8$, $p = 0.006$; NSFT: $t_{(30)} = –2.12$, $p = 0.04$). Given the results following

**Table 1.** Pearson correlation coefficient analysis between TST and FST, TST and OFT, and FST and TST.

The correlation was calculated from data of all mice used in each series of experiments for *Figure 2J–L*, *Figure 6B–D*, or *Figure 7B–D*.

| Figure | Test | Sample numbers | Comparison | Coefficient (r) | p value | Significance |
|--------|------|----------------|------------|-----------------|---------|--------------|
| | | | TST vs FST | 0.464 | $1.17 \times 10^{-7}$ | *** |
| *Figure 2J-L* | Effects of excitation of crus I PCs on RS | 119 | TST vs OFT | 0.153 | 0.0961 | |
| | | | FST vs OFT | 0.0303 | 0.743 | |
| | | | TST vs FST | 0.327 | $3.42 \times 10^{-4}$ | *** |
| *Figure 6B-D* | Effects of inhibition of VTA-projecting DCN neurons on RS | 116 | TST vs OFT | –0.119 | 0.204 | |
| | | | FST vs OFT | 0.0474 | 0.0613 | |
| | | | TST vs FST | 0.439 | $1.43 \times 10^{-4}$ | *** |
| *Figure 7B-D* | Only excitation of VTA-projecting DCN neurons | 70 | TST vs OFT | –0.0308 | 0.8 | |
| | | | FST vs OFT | –0.0143 | 0.907 | |

chemogenetic inhibition of VTA-projecting DCN neurons, this is in line with our data showing a reduction in depression-like behaviors after excitation of inhibitory PCs in crus I. Even though we primarily targeted the AAV injection into the DN, VTA-projecting DCN neurons, not only in the DN, but also in the IPN, might also be chemogenetically inhibited, as the IPN had minor projections to the VTA and is anatomically close to the DN. Taken together, these results indicate that decreased activity of VTA-projecting DCN neurons in the DN, and possibly IPN, inhibits depression-like behaviors.

## Excitation of VTA-projecting DCN neurons is sufficient to trigger depression-like behaviors without stress application

Two possibilities can be considered regarding the involvement of the circuit composed of VTA-projecting DCN neurons in the stress-mediated development of depression-like behaviors, that is the increase in circuit activity proactively promotes the development of depression-like behaviors, or contributes to the development by cooperatively working with other neuronal circuits affected by RS. In the former case, chronic and selective activation of the circuit alone may cause the development of behavioral changes similar to ones induced by stress, even without RS application, whereas it would not trigger such behavioral changes if the latter is the case. To test these possibilities, we performed triple AAV injections into wild-type mice to express Gq in VTA-projecting DCN neurons, as in *Figure 7—figure supplement 1A*. Without applying RS, we then chronically and selectively activated VTA-projecting DCN neurons for 2 weeks by administering CNO once a day to mice expressing Gq in VTA-projecting DCN neurons in the DN (*Figure 7A*, Gq-DN-CNO group). The CNO administration itself without stereotaxic AAV injection did not affect immobility of control mice in the TST and FST, because the CNO group showed similar immobility time to the CTR group, and significantly shorter immobility time than the RS group (*Figure 7—figure supplement 1B and C*; CTR – Sal, TST: $t_{(42)} = 1.1$, p = 0.27, FST: $t_{(42)} = 0.94$, p = 0.35; CTR – CNO, TST: $t_{(42)} = 1.08$, p = 0.28, FST: $t_{(42)} = 0.96$, p = 0.34; RS – Sal, TST: $t_{(42)} = –2.71$, p = 0.008, FST: $t_{(42)} = –2.6$, p = 0.01; RS – CNO, TST: $t_{(42)} = –2.74$, p = 0.008, FST: $t_{(42)} = –2.58$, p = 0.01). These results indicate that off-target effects of CNO, if any, have little or no impact on our behavioral observation without RS application. In contrast, the Gq-DN-CNO group showed a significantly higher immobility compared with the CTR group in both the TST and FST, and the increased immobility time was similar to the RS group (*Figure 7B and C*; CTR – Gq-DN-CNO, TST: $t_{(34)} = 3.31$, p = 0.001, FST: $t_{(34)} = 6.48$, p < 0.001; RS – Gq-DN-CNO, TST: $t_{(33)} = –1.19$, p = 0.24, FST: $t_{(33)} = 0.53$, p = 0.59). Conceivably due to the combination of very mild stress, including daily intraperitoneal administration, the saline (Gq-DN-Sal) group without RS showed significantly longer immobility time than the CTR group in the FST, but not in the TST (TST: $t_{(33)} = 1.34$, p = 0.18, FST: $t_{(33)} = 4.13$, p < 0.001). Importantly, the Gq-DN-CNO group was still immobile for a significantly longer period compared with the Gq-DN-Sal group in both the TST and FST (TST: $t_{(37)} = 2.05$, p = 0.04, FST: $t_{(37)} = 2.4$, p = 0.02). The OFT confirmed that the increased immobility times of the Gq-DN-CNO groups in the TST and FST were not due to less locomotor activity (*Figure 7D*; $F_{(3,66)} = 2.13$, p = 0.11). These results indicate that the chemogenetic excitation of VTA-projecting DCN neurons could result in behavioral consequences similar to the RS.

Nevertheless, there is a possibility that the increased immobility time of the Gq-DN-CNO group may be a presentation of biological events differing from the RS-dependent depression-like behaviors. If this is the case, additive effects could be observed in increased immobility for the Gq-DN-CNO and RS groups. However, the combination of chemogenetic excitation of VTA-projecting DCN neurons with the RS (RS-Gq-DN-CNO group) did not show an additive effect in the TST and FST (*Figure 7—figure supplement 1E and F*). The average immobility time of the RS-Gq-DN-CNO group was significantly longer than that of the CTR group, but was similar to those of the RS group in the TST and FST (CTR – RS-Gq-DN-CNO, TST: $t_{(45)} = 1.99$, p = 0.049, FST: $t_{(45)} = 2.16$, p = 0.03; RS – RS-Gq-DN-CNO, TST: $t_{(39)} = –0.28$, p = 0.78, FST: $t_{(39)} = –1.08$, p = 0.28), while travel distance in the OFT was not different between groups (*Figure 7—figure supplement 1G*; $F_{(3,83)} = 1.28$, p = 0.29). Thus, the increased immobility triggered by the chemogenetic excitation of VTA-projecting DCN neurons alone is likely due to the emergence of depression-like behaviors. This notion was further strengthened by assessing different aspects of depression-like behaviors using the SST and NSFT after 2 weeks of chemogenetic excitation (*Figure 7E*). The Gq-DN-CNO group, having chronic excitation of VTA-projecting DCN neurons alone without the RS, showed significantly less grooming time in the SST and significantly longer latency to feed in the NSFT than the CTR and Gq-DN-Sal groups (*Figure 7F*

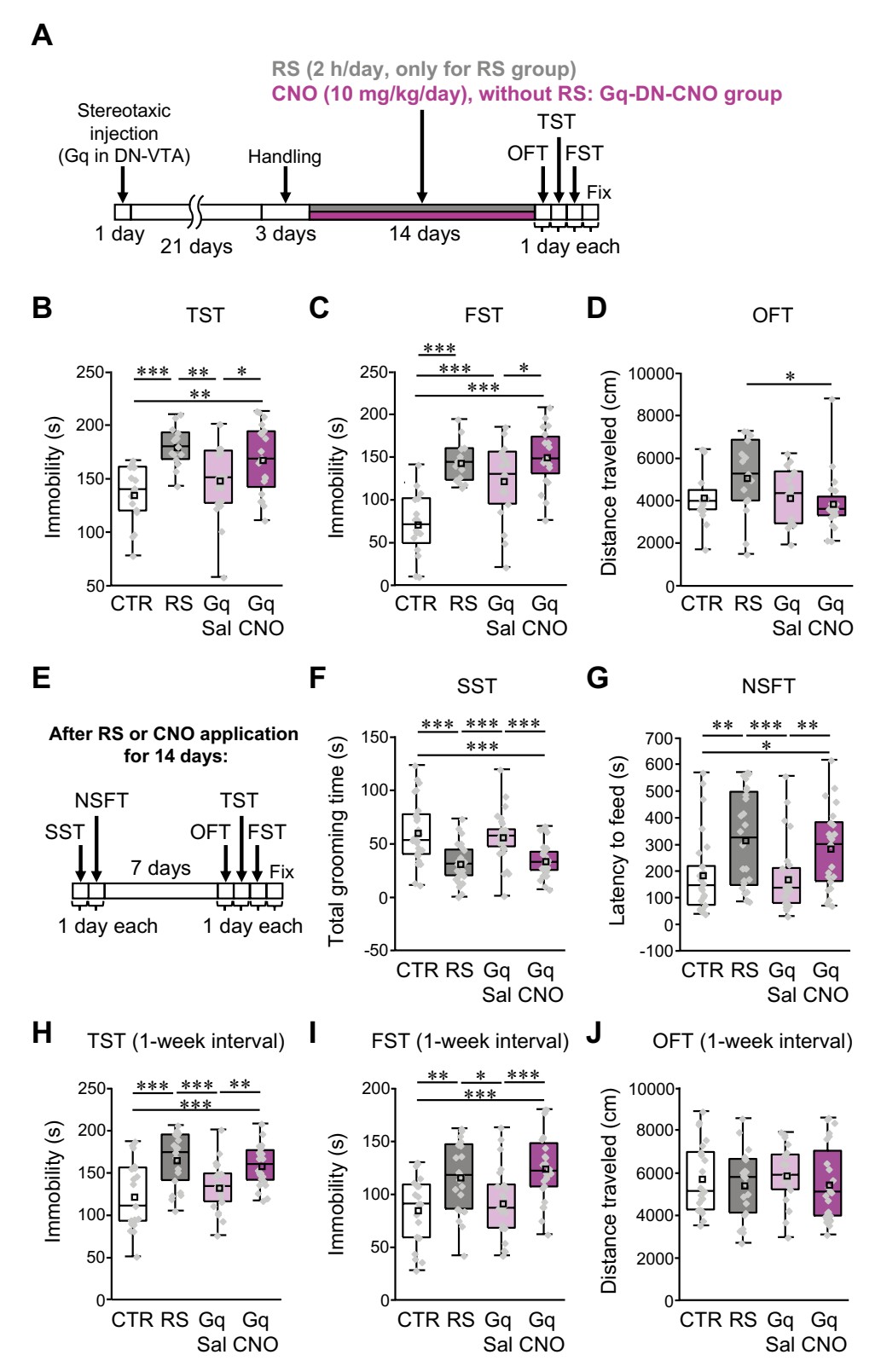

**Figure 7.** Excitation of VTA-projecting DCN neurons in the DN is sufficient to trigger depression-like behaviors in the absence of stress. (**A and E**) Diagram of the experimental time course to check the effects of the excitation of VTA-projecting DCN neurons alone on depression-like behaviors. The OFT, TST, and FST were performed after 2 weeks of chronic administration of CNO into C57BL/6 mice expressing Gq in VTA-projecting DCN neurons

*Figure 7 continued on next page*

*Figure 7 continued*

without RS application (**A**). In the separate series of experiments, the SST and NSFT were performed after 2 weeks of CNO administration, and the OFT, TST, and FST were then performed a week after the last CNO administration (**E**). The combined AAV injection was carried out, as shown in *Figure 7—figure supplement 1A*. (**B–D** and **F–J**) Effects of the chronic excitation of VTA-projecting DCN neurons alone in C57BL/6 mice. The TST (**B**), FST (**C**), and OFT (**D**) were performed by following the time course shown in A (n = 16, 15, 19, and 20 mice for the CTR, RS, Gq-DN-Sal, and Gq-DN-CNO groups, respectively). In the separate series of experiments, the SST (**F**), NSFT (**G**), TST (**H**), FST (**I**), and OFT (**J**) were performed by following the time course shown in **E** (numbers of mice used for the CTR, RS, Gq-DN-Sal, and Gq-DN-CNO groups; n = 27, 27, 29, and 30 mice in the SST; n = 26, 24, 27, and 23 mice in the NSFT; n = 21, 21, 23, and 24 mice in the TST, FST, and OFT). *p < 0.05, **p < 0.01, ***p < 0.001, one-way ANOVA followed by the Fisher's LSD post hoc test. Data are presented as boxplots, as described in the legend to *Figure 1*. Exact p values and the statistical tests used are available in *Figure 7—source data 1*. Effect sizes for behavioral data are available in *Figure 7—source data 2* and n numbers for behavioral tests are available in *Figure 7—source data 3*.

The online version of this article includes the following source data and figure supplement(s) for figure 7:

**Source data 1.** p Values and statistical tests related to *Figure 7*.

**Source data 2.** Effect sizes for behavior results related to *Figure 7*.

**Source data 3.** Numbers of mice for each sex used in each behavioral test related to *Figure 7*.

**Figure supplement 1.** No additive effects of chemogenetic excitation on RS-dependent depression-like behaviors.

**Figure supplement 2.** Changes in body weights of mice used for *Figure 7*.

---

*and G*; CTR – Gq-DN-CNO, SST: $t_{(55)}$ = –4.61, p < 0.001, NSFT: $t_{(47)}$ = 2.29, p = 0.02; Gq-DN-Sal – Gq-DN-CNO, SST: $t_{(57)}$ = –3.94, p < 0.001, NSFT: $t_{(48)}$ = 2.67, p = 0.009), without affecting home-cage feeding consumption (*Figure 7—figure supplement 1H*; $F_{(3,96)}$ = 0.09, p = 0.96). This resulted in the Gq-DN-CNO group being equivalent to the RS group in regard to the grooming time in the SST and the latency to feed in the NSFT (RS – Gq-DN-CNO, SST: $t_{(55)}$ = 0.42, p = 0.68, NSFT: $t_{(45)}$ = –0.7, p = 0.49), implying that the Gq-DN-CNO group developed depression-like behaviors. Furthermore, when the OFT, TST, and FST were performed a week after the last CNO administration (*Figure 7E*), the results were similar to ones observed a few days after the last CNO administration: the immobility time in the Gq-DN-CNO group was significantly longer than the CTR and Gq-DN-Sal groups, but was similar to the RS group in both the TST and FST (*Figure 7H and I*; CTR – Gq-DN-CNO, TST: $t_{(43)}$ = 3.91, p < 0.001, FST: $t_{(43)}$ = 4.01, p < 0.001; Gq-DN-Sal – Gq-DN-CNO, TST: $t_{(45)}$ = 2.84, p = 0.006, FST: $t_{(45)}$ = 3.43, p < 0.001; RS – Gq-DN-CNO, TST: $t_{(43)}$ = –0.72, p = 0.47, FST: $t_{(43)}$ = 0.85, p = 0.4), whereas no difference was detected in the OFT (*Figure 7J*; $F_{(3,85)}$ = 0.43, p = 0.73). The results indicate that the depression-like behaviors triggered by the chemogenetic excitation of VTA-projecting DCN neurons were persistent at least for a week. Taken together with the other results in this study, we conclude that chronic activation of specific DCN neurons projecting from the DN to the VTA is triggered by the chronic stress, and such activation itself proactively leads to the development of depression-like behaviors.

## Discussion

In the present study, to test the role of the cerebellum in the development of depression-like behaviors, chemogenetic manipulation was performed on mice specifically during chronic RS application. Our results demonstrated that chemogenetic excitation of crus I PCs suppressed the activation of DCN neurons in the DN and depression-like behaviors, both of which were triggered by the application of RS. As we identified neuronal networks of the DN connecting from crus I of the cerebellar cortex to the VTA, a brain region involved in the regulation of stress susceptibility (*Fox and Lobo, 2019*), we performed chemogenetic inhibition of VTA-projecting DCN neurons in the DN, and possibly the IPN. We found that the inhibition of these neurons during chronic RS also resulted in the suppression of depression-like behavior. These results indicate that the cerebellum directly contributes to the mechanism that develops depression-like behavior owing to stress, through the activation of a specific circuit composed of DCN neurons projecting to the VTA. In addition, specific activation of the VTA-projecting DCN neurons on their own mimicked the effects of RS on depression-like behavior. We thus

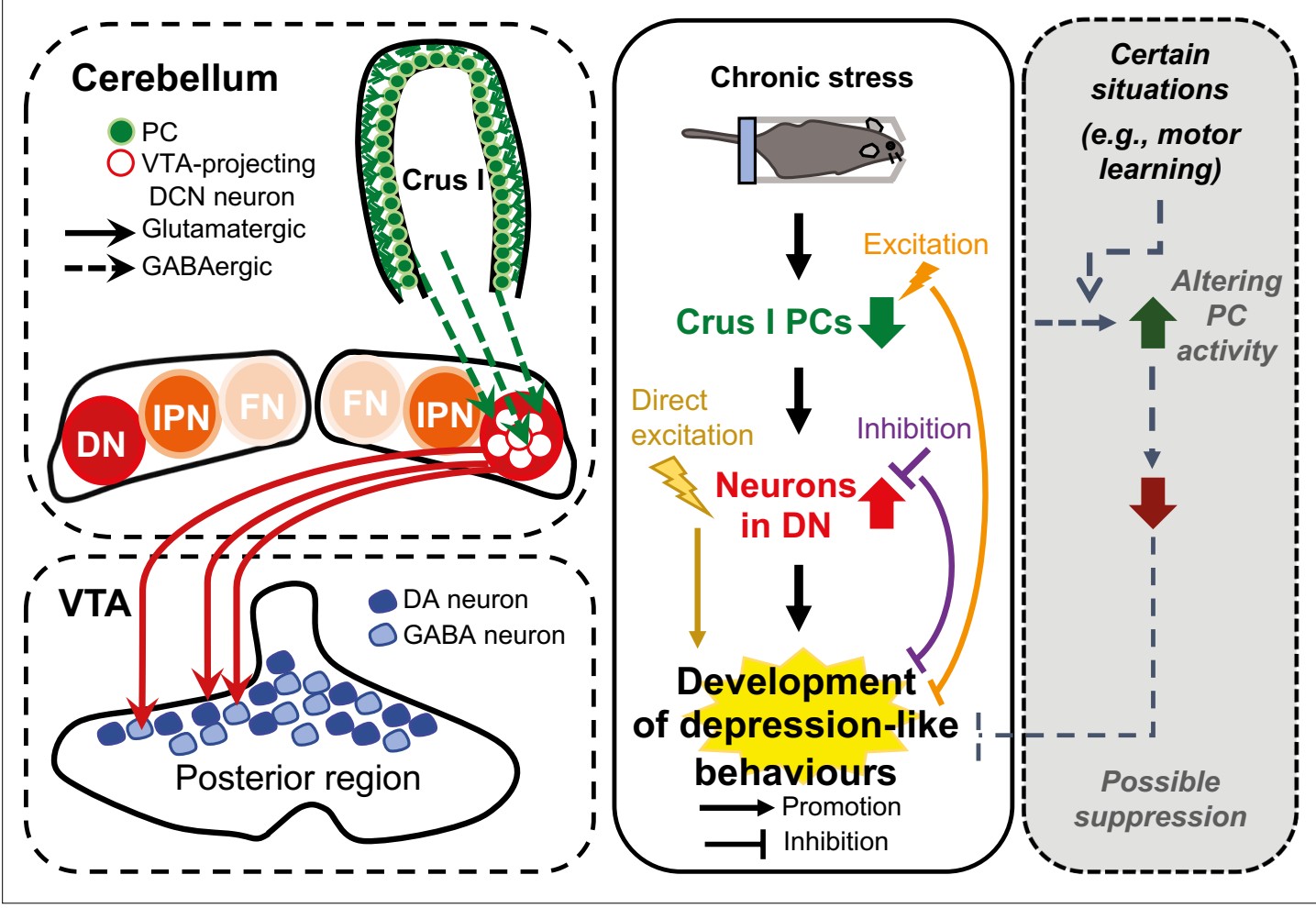

**Figure 8.** Diagram summarizing the results of this study. In this study, we demonstrated that the neural circuit from crus I of the cerebellar cortex to the DN and then to the VTA is functionally connected (left), and that the neuronal circuit composed of VTA-projecting DCN neurons in the DN is crucial for controlling the chronic stress-mediated development of depression-like behaviors (middle), which is possibly prevented under certain situations that may increase PC activity (right).

conclude that the activation of the neuronal circuit of VTA-projecting DCN neurons, which is regulated by crus I PCs, is a crucial determinant of the development of stress-mediated depression-like behaviors (*Figure 8*), although the circuit may include not only a pathway to the VTA but also pathways to non-VTA regions via DCN collateral projections.

## Network connections between the cerebellum and VTA, and their importance for the development of depression-like behaviors

The cerebellum has connections with many brain regions through monosynaptic or multisynaptic pathways (*Bostan and Strick, 2018*; *Bohne et al., 2019*; *Milardi et al., 2019*; *Watson et al., 2019*; *Wagner and Luo, 2020*). The VTA also receives inputs from and sends outputs to many brain regions (*Beckstead, 1978*; *Swanson, 1982*; *Oades and Halliday, 1987*; *Zahm et al., 2011*; *Watabe-Uchida et al., 2012*; *Beier et al., 2015*). Reciprocal anatomical connections between these two brain regions have long been known (*Snider et al., 1976*; *Oades and Halliday, 1987*; *Ikai et al., 1992*). We confirmed the neuronal circuit from the DN and to a lesser extent from the IPN to the VTA, and further clarified functional connections from crus I to the DN and then to the VTA. Whereas all recorded DCN neurons showed IPSCs upon the stimulation of crus I PC axons, 24% of VTA neurons responded to the stimulation of DCN neuronal axons. This seems to be reasonable, considering that individual VTA neurons have selective synaptic connections, as shown by 10–50% of nucleus accumbens-projecting

VTA neurons responding to stimulation of inputs from specific brain regions (*Beier et al., 2015*). Furthermore, we observed zonal organization of corticonuclear and nucleocortical pathways in the VTA-projecting DCN neurons, as is generally detected in DCN neurons (*Houck and Person, 2014*).

Nonmotor functions of the cerebellum are becoming generally accepted (*Hull, 2020*; *Rochefort et al., 2013*; *Wagner and Luo, 2020*), and the cerebellum has also been implicated in stress responses and stress-associated disorders (*De Bellis and Kuchibhatla, 2006*; *Alalade et al., 2011*; *Baldaçara et al., 2011*; *Liu et al., 2012*; *Guo et al., 2012*; *Guo et al., 2013*; *Gounko et al., 2013*; *Phillips et al., 2015*; *Córdova-palomera, 2016*; *Huguet et al., 2017*; *Xu et al., 2017*; *Bambico et al., 2018*; *Depping et al., 2018*; *Moreno-Rius, 2019*). However, the exact role of the cerebellum in stress and depression remained unclear. Antidepressant effects of the cerebellum was suggested by a study in which electrical stimulation of the vermis of stressed animals led to their recovery from depression-like behaviors (*Bambico et al., 2018*). This is likely through the activation of serotonergic neurons in the dorsal raphe nucleus, which were found to have antidepressant effects (*Urban et al., 2016*; *You et al., 2016*; *Nishitani et al., 2019*). In contrast, the proactive contribution of the cerebellum to the development of depression-like behaviors has not been tested. In the present study, we used chemogenetic molecules to manipulate the cerebellar circuit activity during chronic stress application, rather than during or right before testing depression-like behaviors. This experimental design was appropriate for analyzing the direct contribution of the cerebellum to the developmental process of stress-induced depression-like behaviors. Indeed, such manipulation of neuronal activity resulted in the inhibition of stress-mediated depression-like behaviors, indicating the importance of the cerebellum, through the neuronal circuit consisting of VTA-projecting DCN neurons, on the development of depression-like behaviors. Because our results only apply to the development, a possibility of cerebellar contribution to the expression of depression-like behaviors is not precisely ruled out in this study. Further analyses using the manipulation of the circuit only before the behavioral test would resolve this question.

Based on our results of chemogenetic manipulation during RS application, we conclude that the increased activity of VTA-projecting DCN neurons in the DN promotes depression-like behaviors during chronic stress. Given that the chemogenetic activation of crus I PCs reduced depression-like behaviors, the decrease in PC firing is probably the cause of the activity increase in DCN neurons. In fact, a previous study reported that PC firing was decreased by chronic stress application (*Bambico et al., 2018*). In addition, the present study demonstrated that chronic excitation of VTA-projecting DCN neurons in the DN alone triggered depression-like behavior, not only in the TST and FST, but also in the SST and NSFT, to a similar extent as RS application. This leads to the conclusion that the chronic increase in activity of the circuit of VTA-projecting DCN neurons is not only necessary but also sufficient for the development of depression-like behaviors. Considering that depressive symptoms are generally sustained in depressive disorders, this conclusion can be further supported by our results showing sustained depression-like phenotypes in mice (*Figure 7H and I*). Interestingly, excitation of this circuit did not additionally enhance RS-mediated depression-like behavior. Based on the concept of saturation or occlusion that are often used for the investigation of synaptic plasticity as mechanisms of memory formation (*Lisman, 2017*; *Inoshita and Hirano, 2018*), the stimuli will not have additive effects if two types of stimuli triggering similar consequences are mediated by the same pathway. Therefore, our results showing no additive effects suggest that RS-induced depression-like behavior is mediated by the activation of a circuit consisting of VTA-projecting DCN neurons.

Because this circuit obviously includes a direct pathway from the DN to the VTA, which is the primary target of AAV-mediated chemogenetic molecule expression, activity of the DN-VTA circuit is likely the crucial determinant of stress-mediated depression-like behaviors. To a lesser extent, our tracing analyses showed the neuronal circuit also from the IPN to the VTA, so that the IPN-VTA circuit could be a part of the determinant of depression-like behaviors as well. There may also be the contribution of DCN neuron axon collateral pathways, as DCN neurons showed broad projection patterns (*Kebschull et al., 2020*). We have found the nucleocortical pathway of VTA-projecting DCN neurons. A previous study of associative motor learning demonstrated that the activity of nucleocortical projections from the IPN contributes to the amplification of learned response (*Gao et al., 2016*). In this previous study, the optogenetic activation of nucleocortical pathway increased the learned response in trained animals, yet did not trigger the response in untrained animals (*Gao et al., 2016*). As we found that the activity of VTA-projecting DCN neurons led to the depression-like behaviors, their nucleocortical pathway may also be involved in promoting the development of depression-like behaviors.

However, given the modulatory effects of nucleocortical pathway shown by Gao et al., depression-like behaviors triggered by chronic excitation of VTA-projecting DCN neurons alone would not only be due to the activation of nucleocortical pathway, but possibly due to the nucleocortical pathway amplifying responses triggered by the efferent pathway. In addition, because GFP signals were still weakly visible in the RN when VTA-projecting DCN neurons were selectively labeled (*Figures 3G and 5B*), their collateral axons may partly project to the RN. The cerebellar-RN pathway is well known for cerebellum-dependent associative motor learning, delay eyeblink conditioning (*Freeman and Steinmetz, 2011*). Whereas there has been an idea from evolutionary point of view that projections from the DN to the parvocellular part of the RN might be involved in cognitive functions (*Basile et al., 2021*), their relevance to stress-related responses is totally unknown. It is therefore unclear whether or how the collateral projections to the RN is involved in the stress-dependent behavioral changes we observed. Among all circuits of VTA-projecting DCN neurons, the precise circuit involved in the regulation of depression-like behaviors might be clarified by further studies using local neuronal activation by optogenetic stimulation, although it may be challenging to chronically manipulate neuronal activity for several hours each day.

## Mechanisms of the cerebellum regulating depression-like behaviors

The present study demonstrated for the first time that the cerebellum proactively mediate chronic stress-induced depression-like behaviors via the circuit composed of VTA-projecting DCN neurons. This finding consequently leads to two important questions that should be addressed in the near future, and we therefore discuss the possibility regarding these questions. First, considering that chronic stress triggers sustained changes in behaviors, certain forms of plasticity likely occur in the circuit mediating such changes. As many forms of plasticity have been reported in the cerebellum (*D'Angelo et al., 2016*; *Ohtsuki et al., 2020*), the plasticity of synapses or intrinsic excitability (IE) may be involved in the process of stress application leading to the decrease in PC firing and the increase in DCN neuron activity. A sequence of plasticity suggested by a study of motor memory (*Jang et al., 2020*) could be applicable to this process: long-term depression of IE is triggered in crus I PCs during RS application, and long-term potentiation of IE in VTA-projecting DCN neurons of the DN is consequently induced, which could lead to repeated activation of these DCN neurons during chronic RS application. Further, given that several forms of plasticity in the VTA can be triggered by chronic stress and determine stress susceptibility (*Douma and de Kloet, 2020*), the repeated DN-VTA circuit activity would regulate some forms of plasticity in the VTA, resulting in the development of depression-like behaviors. Thus, understanding the natures of plasticity not only within the cerebellum but also in the DN-VTA circuit seems to be an interesting follow-up study.

The second interesting question is as to what kind of signals are processed through the cerebellum and the circuit of VTA-projecting DCN neurons. In contrast to our results, a previous study demonstrated that activation of the circuit from the DCN to the VTA is rewarding (*Carta et al., 2019*). To reconcile our results with this previous study, two different experimental conditions are important to be considered. First, to manipulate the circuit, we used chemogenetics, whereas the previous study used optogenetics. Second, we chronically manipulated VTA-projecting DCN neurons for 2 weeks during stress application before the behavioral tests, whereas the previous study mostly manipulated DCN-VTA pathway under the behavioral recordings. Both combinations of experimental conditions appear to be reasonable from the time perspective, namely, chemogenetics with a low temporal resolution for chronically repeated manipulations and optogenetics with a high temporal resolution for transient manipulations. Although other possibilities cannot be denied, such temporal aspects, including the timing, duration and frequency, for manipulation of the DCN-VTA pathway could be the major factors leading to different consequences in these two studies. As the DA neurons in the VTA have been implicated in the regulation of both mentally positive and negative events, such as reward and aversion, respectively (*Volman et al., 2013*; *Pignatelli and Bonci, 2015*), the promotion of both reward and depression-like behavior via temporally different activity patterns in the circuit composed of VTA-projecting DCN neurons appear to be justified. The cerebellar functions operating prediction of not only motor, but also non-motor events, have drawn attention (*Sokolov et al., 2017*; *Hull, 2020*). In line with this notion, signals related not only to the reward prediction, but also to the pessimistic prediction, may be provided via VTA-projecting DCN neurons. Further investigations performed in consideration of coordinating the abovementioned differences between two studies

may provide better understanding of signals processed through the circuit of VTA-projecting DCN neurons.

## Possible involvement of the cerebellum in the resilience to mental disorders

The inhibition of depression-like behavior by chemogenetic PC excitation that was observed in our study is likely to occur through inhibition of the increased activity of DCN neurons, which is a cause of the development of depression-like behavior. In previous studies, observations of movement behaviors together with recording of PC firing have shown that PC firing rates correlate with the kinematics of movement (*Medina, 2011*; *Brown and Raman, 2018*; *Popa et al., 2019*). Besides, PC firing patterns have been shown to change in parallel with learned movement (*Jirenhed et al., 2007*). Enhancement of the intrinsic excitability of PCs was also observed upon associative motor learning (*Titley et al., 2020*). These observations indicate that PC activity can be increased in a wide variety of situations. Therefore, even though chronic stressful events may cause a reduction in PC activity, depression-like behaviors can be prevented depending on the situations that lead to an increase in PC activity (*Figure 8*, right panel), as we found by the chemogenetic excitation of PCs. While a number of factors in many brain regions are known to control an individual's susceptibility and resilience to stress (*Franklin et al., 2012*; *Osório et al., 2017*; *Knowland and Lim, 2018*; *Liu et al., 2018*), the cerebellum did not previously draw much attention as one of these brain regions. To understand whether and how the cerebellum contributes to stress resilience depending on the situation, it would be beneficial to clarify the superficially conflicting aspects of cerebellar network structures, the compartmentalization of the cerebellum into functional modules (*D'Angelo and Casali, 2013*; *Apps et al., 2018*), and the structures appropriate for input integration (*Huang et al., 2013*; *Ishikawa et al., 2015*).

We found that alterations of activities in the cerebellum and likely in the circuit to the VTA by chronic stress is a cause of depression-like behaviors. This finding supports the notion in human that cerebellar abnormalities that correlate with the symptoms of mental disorders, including depression (*Alalade et al., 2011*; *Córdova-palomera, 2016*; *Meabon et al., 2016*; *Xu et al., 2017*; *Romer et al., 2018*; *Moberget et al., 2019*), are not just an outcome indicating the degree of symptoms, but are risk factors that predict the possibility of experiencing mental disorders in the future (*Romer et al., 2018*; *Hariri, 2019*; *Moberget et al., 2019*). In addition to the VTA, the cerebellum has interactions through multisynaptic pathways with other brain regions implicated in the regulation of mood, such as the medial prefrontal cortex, dorsal raphe nucleus, and hippocampus (*Weiss and Pellet, 1982*; *Braz et al., 2009*; *Chen et al., 2016*; *Watson et al., 2014*; *Watson et al., 2019*; *Bohne et al., 2019*), and these interactions might cooperatively work in controlling psychological conditions. Nevertheless, the present study demonstrated that the circuit composed of VTA-projecting cerebellar neurons is a crucial pathway in controlling stress-mediated depression-like behaviors. This raises the possibility that maintaining the integrity and functionality of the cerebellum, particularly the crus I and the DN, may be important in regulating neural activities in the VTA and other mood-related brain regions, and consequently in avoiding the development of mental depression in human as well.

## Materials and methods

### Key resources table

| Reagent type (species) or resource | Designation | Source or reference | Identifiers | Additional information |
|---|---|---|---|---|
| Strain, strain background (*Mus musculus*) | C57BL/6 J mice | Orient Bio | N/A | |
| Strain, strain background (*Mus musculus*) | ICR mice; CrljOri:CD1 | Orient Bio | N/A | |
| Genetic reagent (*Mus musculus*) | *Gad2*-IRES-Cre mice; Gad2tm2(cre)Zjh/J | Jackson Laboratories | Stock No: 010802; RRID:IMSR_JAX:010802 | |
| Genetic reagent (*Mus musculus*) | Pcp2-cre mice; B6.129-Tg(Pcp2-cre)2Mppin/J | Jackson Laboratories | Stock No: 004146; RRID:IMSR_JAX:004146 | |
| Genetic reagent (*Mus musculus*) | Ai six reporter mice; B6.Cg-Gt(ROSA)26Sortm6(CAG-ZsGreen1)Hze/J | Jackson Laboratories | Stock No: 007906; RRID:IMSR_JAX:007906 | |

*Continued on next page*

*Continued*

| Reagent type (species) or resource | Designation | Source or reference | Identifiers | Additional information |
|---|---|---|---|---|
| Antibody | Mouse monoclonal anti-calbindin | Sigma-Aldrich | Cat#C9848; RRID:AB_476894 | (1:200) |
| Antibody | Rabbit polyclonal anti-TH | Millipore | Cat#AB152; RRID: AB_390204 | (1:400) |
| Antibody | Rabbit monoclonal anti-c-Fos | Cell Signalling Technology | Cat#2250; RRID: AB_2247211 | (1:300) |
| Antibody | Mouse monoclonal anti-NeuN | Millipore | Cat#MAB377; RRID: AB_2298772 | (1:200) |
| Antibody | Mouse monoclonal anti-GAD67 | Millipore | Cat#MAB5406; RRID:AB_2278725 | (1:1000) |
| Antibody | Mouse monoclonal anti-GFAP | Millipore | Cat#MAB360; RRID:11212597 | (1:200) |
| Antibody | Goat anti-mouse IgG(H + L) alexa fluor@647 | Invitrogen | Cat#A21235; RRID: AB_2535804 | (1:200) |
| Antibody | Goat anti-rabbit IgG(H + L) alexa fluor@647 | Invitrogen | Cat#A21245; RRID: AB_141775 | (1:200) |
| Antibody | Rabbit polyclonal anti-OLLAS | Antibodies-online | ABIN1842163 | (1:1000) |
| Chemical compound, drug | Clozapine N-oxide | Hello Bio | Cat#HB1807 | 10 mg/kg, i.p. |
| Chemical compound, drug | Bicuculline methochloride | Tocris Bioscience | Cat#0131 | |
| Chemical compound, drug | Mounting medium | Vector Laboratories | Cat#H-1400 | |
| Chemical compound, drug | Mounting medium with DAPI | Vector Laboratories | Cat#H-1500 | |
| Chemical compound, drug | Prolong diamond antifade mountant | Thermo Fisher Scientific | Cat#P36961 | |
| Chemical compound, drug | Cholera toxin subunit B, Alexa flour 647 conjugate | Thermo Fisher Scientific | Cat#C34778 | |
| Chemical compound, drug | Tetrodotoxin | Tocris Bioscience | Cat#1,069 | |
| Chemical compound, drug | 4-Aminopyridine | Tocris Bioscience | Cat#0940 | |
| Recombinant DNA reagent | pAAV-hSyn-DIO-hM3Dq(Gq)-mCherry | Addgene | Addgene plasmid #44,361 | |
| Recombinant DNA reagent | pAAV-hSyn-DIO-hM4Di(Gi)-mCherry | Addgene | Addgene plasmid #44,362 | |
| Recombinant DNA reagent | pAAV-CaMKIIa-hChR2(H134R)-EYFP | Addgene | Addgene plasmid #26,969 | |
| Recombinant DNA reagent | AAV-CaMKIIa-GCaMP6f-P2A-nls-dTomato | Addgene | Addgene plasmid #51,087 | |
| Recombinant DNA reagent | pBAD-mTagBFP2 | Addgene | Addgene plasmid #34,632 | |
| Recombinant DNA reagent | ptdTomato-N1 | Clontech | Cat#632,532 | |
| Recombinant DNA reagent | paavCAG-iCre | Addgene | Addgene plasmid #51,904 | |
| Recombinant DNA reagent | paavCAG-sfGFP | Addgene | Will be deposited | |
| Recombinant DNA reagent | paavCAG-pre-mGRASPi-mCerulean | Addgene | Will be deposited | |
| Recombinant DNA reagent | paavCAG-post-mGRASPi-2AdT | Addgene | Addgene plasmid #34,912 | |
| Recombinant DNA reagent | rAAV2-retro helper | Addgene | Addgene plasmid #81,070 | |
| Biological sample (AAV) | AAV-sSyn-GFP | Kim et al., *Kim et al., 2015a* | | |
| Biological sample (AAV) | AAV-sSyn-tdT | This paper | | Further information will be provided by the Lead Contact upon request. |
| Biological sample (AAV) | AAV-sSyn-mTBFP | This paper | | Further information will be provided by the Lead Contact upon request. |

*Continued on next page*

*Continued*

| Reagent type (species) or resource | Designation | Source or reference | Identifiers | Additional information |
|---|---|---|---|---|
| Biological sample (AAV) | AAV-sSyn-ChR2-GFP | This paper | | Further information will be provided by the Lead Contact upon request. |
| Biological sample (AAV) | AAV-sSyn-ChR2-2AdT | This paper | | Further information will be provided by the Lead Contact upon request. |
| Biological sample (AAV) | AAV-sSyn-FLEX-GFP | This paper | | Further information will be provided by the Lead Contact upon request. |
| Biological sample (AAV) | AAV-sSyn-FLEX-ChR2-GFP | This paper | | Further information will be provided by the Lead Contact upon request. |
| Biological sample (AAV) | AAV-sSyn-FLEX-hM3Dq(Gq)–2AGFP | This paper | | Further information will be provided by the Lead Contact upon request. |
| Biological sample (AAV) | AAV-sSyn-FLEX-hM4Di(Gi)–2AGFP | This paper | | Further information will be provided by the Lead Contact upon request. |
| Software, algorithm | pClamp 10 | Axon instruments | RRID:SCR_011323 | |
| Software, algorithm | NIS-Element | Nikon | RRID:SCR_014329 | |
| Software, algorithm | Fiji | NIH | RRID:SCR_002285 | |
| Software, algorithm | Ethovision | Noldus | RRID:SCR_000441 | |
| Software, algorithm | Origin | Origin Lab | RRID:SCR_014212 | |

## Mice

All experiments were performed in accordance with the Institutional Animal Care and Use Committee of Korea Institute of Science and Technology. ICR mice were used for neuronal tracing analysis, and C57BL/6 J and PCP2-Cre transgenic mice (Jackson Laboratories, B6.129-Tg(Pcp2-cre)2Mppin/J) were used for behavioral and c-Fos immunohistochemical analyses. For the electrophysiological experiments, C57BL/6 J, PCP2-Cre, or GAD2-IRES-Cre;Ai6 mice were used. Regarding the PCP2-Cre mice, male PCP2-Cre mice were crossed with female ICR mice to obtain heterozygous PCP2-Cre (PCP2-Cre;ICR) mice for use in this study. GAD2-IRES-Cre (Jackson Laboratories, B6J.Cg-Gad2tm2(cre) Zjh/MwarJ) and Ai6 ZsGreen reporter mice (B6.Cg-Gt(ROSA)26Sortm6(CAG-ZsGreen1)Hze/J) were crossed to obtain GAD2-IRES-Cre;Ai6 mice. Animals of both sexes were used in every set of experiments, and the numbers of mice for each sex used in a single condition of behavioral experiments are listed in all source data three related to *Figure 2—source data 3*, *Figure 6—source data 3*, *Figure 7—source data 3*. Mice were group housed (6 mice/cage) at 23°C to 25 °C under a 12 hr light/12 hr dark cycle with ad libitum access to food and water. The ages of the mice are described in the individual sections explaining the different procedures.

## AAV production and stereotaxic injection

For all experiments, AAV serotype I or rAAV2-retro (*Tervo et al., 2016*) was used for anterograde or retrograde labeling, respectively. AAV constructs were made by cloning using plasmids for AAV-sSyn and AAV-sSyn-FLEX that were used in our previous study (*Kim et al., 2015a*). The cDNA fragments for hM3D (Gq), hM4D (Gi), ChR2, P2A, mTagBFP2, and tdT were obtained from pAAV-hSyn-DIO-hM3D(Gq)-mCherry (Addgene), pAAV-hSyn-DIO-hM4D(Gi)-mCherry (Addgene), pAAV-CaMKIIa-hChR2(H134R)-EYFP (Addgene), AAV-CaMKIIa-GCaMP6f-P2A-nls-dTomato (Addgene), pBAD-mTagBFP2 (Addgene), and ptdTomato-N1 (Clontech), respectively. The plasmids of paavCAG-iCre (Addgene), paavCAG-sfGFP, and paavCAG-post-mGRASP-2AdT (Addgene) made in a previous study (*Druckmann et al., 2014*) were used. The plasmid paavCAG-pre-mGRASPi(OLLAS)–2AiCre was modified for efficient reconstitution and axonal visualization. AAV vectors with estimated titers from $10^{12}$–$10^{13}$ vector genome copies were produced, as described previously (*Kim et al., 2011*; *Kim et al., 2015a*).

For stereotaxic injections, mice were anesthetized with Avertin (250 µg/g body weight) and injected using a stereotaxic apparatus (Narishige) together with a microinjection pump (Nanoliter 2010, WPI, Inc) or iontophoresis (Stoelting). When a confined volume of AAV vectors were injected for neuronal tracing analysis, as shown in *Figure 3*, iontophoretic injection was used (*Oh et al., 2014*), whereas

pressure injection with a microinjection pump was used for the other experiments. The volume of AAV vectors injected by the pressure injection was 0.7 μL in crus I, 0.5 μL in the DCN, and 0.4 μL in the VTA. The conditions used for iontophoretic injection were 5 μA, 7 s on/7 s off cycle, for 5 min. Stereotaxic coordinates of crus I were anteroposterior (AP) −6.3 mm, mediolateral (ML) ±2.75 mm, and ventral (V) −1.30 to −1.50 mm relative to bregma; those of the DCN were AP −6.00 mm, ML ±2.30, and V −2.70 mm relative to bregma; and those of the VTA were AP −3.1 mm, ML 0 to −0.5 mm and V −4.50 mm relative to bregma. The age of the mice subjected to the AAV injections was postnatal day 18 for electrophysiological and neuronal tracing analyses, and 5 weeks old for behavior and c-Fos immunohistochemical analyses. AAV vectors used for the individual analyses are described in the corresponding sections of the results, and are also shown in the figures. For the specific expression of molecules in VTA-projecting DCN neurons, we injected retrogradely infecting AAV expressing Cre recombinase (rAAV2-retro-CAG-iCre) into the VTA, and AAV triggering Cre-dependent expression into the DCN. To confirm the areas of injection in these experiments, AAV-sSyn-tdT (1/4 volume of total) and AAV-sSyn-mTBFP (1/4 volume) were combined with AAV injected into the DCN and VTA, respectively. Similarly, to confirm the injection of rAAV2-retro-CAG-sfGFP into the VTA, AAV-sSyn-mTBFP (1/4 volume) was combined. For electrophysiological analysis on VTA-projecting DCN neurons, the retrograde tracer, CTB 647 (1 mg/ml, Thermo Fisher Scientific), was injected into the VTA. For the analysis of structural synaptic connections, we used mGRASPi, which is based on functional complementation of two non-fluorescent split-GFP fragments targeted specifically to the pre- and post-synaptic membrane that are reconstituted as fluorescent GFP in that location, when two neurons, each expressing one of the fragments, are closely opposed across a synaptic cleft. For this technique, AAV-CAG-pre-mGRASPi(OLLAS)–2AiCre was injected into the DCN and AAV-CAG-post-mGRASPi-2AdT was injected into the VTA. After the surgery, mice were kept on a heating pad until they recovered from the anesthesia and were returned to their home cages.

## Immunohistochemistry and confocal imaging

Approximately 2 weeks after AAV injection for neuronal tracing analysis, 5-week-old mice were anesthetized with isoflurane and perfused transcardially with 4% paraformaldehyde (PFA) in 0.1 M sodium phosphate buffer (pH 7.4). To confirm successful AAV injection in mice subjected to behavior analysis, or to observe c-Fos expression, 10–12 week-old mice were also similarly perfused after all behavior tests or right after 2 weeks of RS, respectively. For the time course of c-Fos expression, mice were perfused right after 3, 7, or 10 days of RS. The postfixation in 2% PFA solution lasted for 16 hr at 4 °C. Brain samples were sliced (100 μm) using a vibratome (Leica VT1200S) in the coronal plane and were blocked for 30 min at 4°C in 5% normal goat serum for all experiments. For calbindin (mouse anti-calbindin, Sigma-Aldrich, C9848) and TH (rabbit anti-TH, Millipore, AB152) staining, slices were incubated with the calbindin antibody (1:200) or TH antibody (1:400) overnight at 4 °C and then incubated with a secondary antibody (1:200, Alexa Fluor 647-conjugated anti-mouse or anti-rabbit IgG antibody, Invitrogen) for 4 hr at room temperature. Cerebellar slices were stained with calbindin to label PCs, and VTA-containing slices were stained with TH to label dopaminergic neurons. For c-Fos staining of DCN neurons, cerebellar slices containing DCN were incubated with a c-Fos antibody (1:300, rabbit, Cell Signaling, 2250 S) for 48 hr and subsequently incubated with secondary antibody (1:200) for 24 hr at 4 °C. We note that we used a concentration of 1:300 of c-Fos antibody by replicating a previous study that demonstrated the increased c-Fos expression in the DCN of rats after chronic stress application (*Huguet et al., 2017*), even though it is not the same c-Fos antibody as ours. Despite the manufacturer instruction from Cell Signaling suggests to use lower concentrations in immunohistochemistry, for example approximately in 1:10,000, its c-Fos antibody has been often used at concentrations varying from 1:100 to 1:1000 (e.g. *Chen et al., 2021*; *Han et al., 2021*; *Phua et al., 2021*; *Trojanowski et al., 2021*), so that the concentration of 1:300 could be justified. For co-staining of DCN neurons for c-Fos with other antibodies, cerebellar slices were firstly stained with c-Fos antibody as mentioned above and then stained with either NeuN (1:200, mouse, Millipore, MAB377), GAD67 (1:1000, mouse, Millipore, MAB5406) or GFAP (1:200, mouse, Millipore, MAB360) antibodies for overnight at 4 °C, which were then incubated with secondary antibody as mentioned above. Slices were then mounted with either DAPI mounting medium (Vector Laboratories, H-1500) or normal mounting solution (Vectashield hard set mounting medium, Vector Laboratories, H-1400).

All the slices stained with NeuN, GAD67, GFAP, calbindin, TH, or c-Fos antibodies were imaged using an A1R laser-scanning confocal microscope (Nikon). Images of whole cerebellar or midbrain slices were taken by scanning a large image using NIS Elements software (Nikon). To analyze c-Fos-positive DCN neurons, high magnified (211.7 × 211.7 μm) z-stack images of c-Fos and DAPI were acquired. All DCN cells labeled with DAPI were detected by 3D objective counter functions of Fiji software (National Institutes of Health), and percentages of the c-Fos-positive neurons were calculated. The percentage increase in c-Fos-positive cells over CTR group was calculated as follows: $((c\text{-}Fos_{RS\text{-}x} - c\text{-}Fos_{CTR\text{-}x})/c\text{-}Fos_{CTR\text{-}x}) \times 100$ (%), where $c\text{-}Fos_{RS\text{-}x}$ stands for percentages of c-Fos-positive cells in DAPI-positive cells in mice receiving x days of RS application, and $c\text{-}Fos_{CTR\text{-}x}$ stands for those in control mice that were simultaneously analyzed. The average numbers of cells counted were summarized in *Table 2*. To screen inputs from FN, IPN, and DN, z-stack images in the VTA areas were acquired. MFTs of VTA-projecting DCN neurons in the cerebellar cortex were acquired as follows: the location of MFT-like structures was first identified by enlarging low magnified images of entire cerebellar slices, and then high-magnification objectives were used to acquire z-stack images of MFTs. The VTA regions were visualized by staining with a TH antibody.

For the analysis of synaptic connections using mGRASPi, mice were anesthetized with isoflurane and perfused transcardially with 4% PFA in 0.1 M sodium phosphate buffer (pH 7.4), 4 weeks after injections of AAV-CAG-pre-mGRASPi(OLLAS)–2AiCre into the DCN and AAV-CAG-post-mGRASPi-2AdT into the VTA at P18. Brain samples were then postfixed in 4% PFA solution at 4 °C for 4 hr. Midbrain slices were made (100 μm) and mounted with mounting solution (Prolong Diamond Antifade Mountant, Thermo Fisher Scientific). VTA areas of the midbrain slices were imaged using an LSM 780 confocal microscope (Zeiss) 4–7 days after the mounting, as previously described (*Feng et al., 2014*).

## Electrophysiology and optogenetic stimulation

### Acute slice preparations

Mice were deeply anesthetized with isoflurane and decapitated. Coronal midbrain slices (250 μm, 6–10 week-old mice) containing VTA or sagittal cerebellar slices (250 μm, P28-P35 mice) containing DN were prepared with warm (about 36 °C) glycerol-based artificial cerebrospinal fluid (aCSF) containing (in mM) 11 glucose, 250 glycerol, 25 $NaHCO_3$, 2.5 KCl, 1.25 $NaH_2PO_4$, 0.05 $CaCl_2$, and 1.3 $MgCl_2$ (oxygenated with 95% $O_2$/5% $CO_2$) (*Ankri et al., 2014*; *Ye et al., 2006*). The temperature of the slicing chamber (36 °C) was maintained constant by pouring hot water into the external chamber. The obtained slices were incubated for an hour in warmed (36 °C) normal aCSF containing (in mM): 125 NaCl, 25 $NaHCO_3$, 2.5 KCl, 1.25 $NaH_2PO_4$, 11 glucose, 1.3 $MgCl_2$, and 2.5 $CaCl_2$. After another 30 min of incubation at room temperature, slices were transferred to a recording chamber and perfused continuously with oxygenated normal aCSF.

### Loose cell-attached and whole-cell patch clamp recording

For loose cell-attached patch clamp recording from PCs, patch pipettes were pulled (4–5 MΩ) and filled with (in mM): 130 potassium gluconate, 2 NaCl, 4 $MgCl_2$, 4 $Na_2$-ATP, 0.4 Na-GTP, 20 HEPES (pH 7.2), and 0.25 EGTA. Spontaneous firing was recorded from the Gq-expressed PCs that were visualized by simultaneously expressed GFP fluorescence using a confocal microscope (Olympus FV1000). To confirm the effects of activation of Gq by CNO (Hello Bio), CNO (10 μM) was added to the perfused aCSF and changes in the firing rate were measured. After the recording during CNO treatment, the slice was washed out with normal aCSF again and the recovery of the firing was monitored. The firing rates recorded were then normalized to the averaged baseline.

For whole-cell patch-clamp recording, patch pipettes were pulled (6–7 MΩ) and filled with the same potassium gluconate internal solution as described above. Light-evoked EPSCs were measured from VTA neurons at a holding potential of −70 mV. VTA neurons were visualized directly using an Olympus BX61WI microscope. Both DA and GABA neurons in the VTA recorded were mainly located in the dorsolateral part where ChR2-positive axons coming from the DN are present. To determine the cell type of the VTA neurons, $I_h$ currents were measured by applying varied negative voltage steps (from −110 to −70 mV) from a holding potential of −60 mV after recording EPSCs. The absence of $I_h$ currents in GABA neurons was confirmed by recording $I_h$ currents in ZsGreen-negative or ZsGreen-positive VTA neurons of GAD2-IRES-Cre;Ai6 mice, which were identified by observation of ZsGreen fluorescence using an Olympus confocal microscope. To measure light-evoked IPSCs, whole-cell

**Table 2.** Numbers of DAPI- and c-Fos-positive cells in each subregion of DCN at different time periods of RS. Numbers counted in single z-stack images are presented. Data are shown as the mean ± SD, and related to *Figures 1 and 2*.

| RS duration | DCN sub-region | Control | | | Stress | | | RS-Gq-CNO | | |
|---|---|---|---|---|---|---|---|---|---|---|
| | | DAPI | c-Fos | N | DAPI | c-Fos | N | DAPI | c-Fos | N |
| 3 day | DN | 94.6 ± 8 | 31.4 ± 4.2 | 5 | 97.2 ± 28.5 | 50.8 ± 14.6 | 5 | - | - | - |
| 7 day | DN | 152 ± 15 | 41.8 ± 14.2 | 5 | 152 ± 21.7 | 96.4 ± 20.2 | 5 | - | - | - |
| 10 day | DN | 128.2 ± 18.7 | 32.6 ± 8.7 | 5 | 119.6 ± 20.3 | 67 ± 11.2 | 5 | - | - | - |
| | DN | 147.6 ± 46.5 | 36.4 ± 12.9 | 8 | 109.3 ± 53.9 | 54.8 ± 22.3 | 8 | 95.8 ± 11.1 | 21.4 ± 10.7 | 5 |
| | IPN | 114.3 ± 30.5 | 35.5 ± 9.5 | 8 | 99.6 ± 32 | 47 ± 24 | 8 | - | - | - |
| 14 day | FN | 103.8 ± 29 | 35.5 ± 9.9 | 8 | 112.1 ± 33.2 | 47.5 ± 21 | 8 | - | - | - |

patch-clamp recordings were made on CTB 647-positive DCN neurons in the DN that were visualized by Alexa Fluor 647 fluorescence using a confocal microscope, with the membrane potential held at −50 mV. To confirm GABA type A receptor-mediated IPSCs, 10 µM of bicuculline methochloride (Tocris Bioscience) was added to the perfused aCSF. For the experiments recording light-evoked EPSCs or IPSCs, we used midbrain or cerebellar slices with numerous ChR2-positive axons visualized by GFP or dT fluorescence in the VTA or the DN. The monosynaptic connections from the DCN to the VTA were confirmed by using TTX (0.5 µM, Tocris Bioscience) and 4-AP (1 mM, Tocris Bioscience), as was done previously (*Carta et al., 2019*; *Yan et al., 2019*). All the electrophysiological recordings were made using MultiClamp 700B amplifier, and data were acquired using pCLAMP software (Molecular Devices). Data were accepted if the input membrane resistance was greater than 100 MΩ, and the holding current was less than 100 pA. Analyses of electrophysiological parameters were performed using OriginPro (OriginLab) and Clampfit (Molecular Devices) software. To summarize the recording results, we measured peak amplitudes of EPSCs and IPSCs evoked by the first photostimulation. Durations between photostimulation and the peak EPSC were also calculated. The amplitude of $I_h$ currents was calculated as the difference between the maximum and minimum values during 500ms of negative voltage steps at −110 mV (*Figure 4—figure supplement 1B*).

To timely apply photostimulation onto ChR2-positive axons of DCN neurons in the VTA or of PCs in the DN, a shutter driver (VCM-Di, Uniblitz, Vincent Associates) was installed in the light path from the fluorescence lamp housing to the microscope. The mechanical shutter was controlled by pCLAMP software, which in turn controls the frequency and exposure duration of the light stimulation (470–495 nm via a band path filter) with a light intensity of 2 mW/mm². To evoke EPSCs or IPSCs, 5 trains of photostimuli were applied at a frequency of 10 or 50 Hz.

## Chronic restraint stress

To deliver restraint stress, 8- to 9-week-old of PCP2-Cre or C57BL/6 J mice were each placed in a well-ventilated 50 mL conical tube with holes along the side and at the tip, where the nose is positioned. Depending on the size of the mouse, a 3.5–7 cm-long tube (15 mL) was inserted into the 50 mL tube to fill the leftover space after the mouse is placed into the 50 mL tube. The 50 mL tube was closed with a cap that had a hole for letting the tail out. Mice were unable to move both forward and backward in the tube. Mice were subjected to 2 hr of this RS every day for 2 weeks, which is the protocol of RS in this study. To enable the comparison within a series of behavioral experiments even in the presence of mild variability between different series, all series of concurrently processed behavioral experiments included both the CTR group and the RS group without expression of chemogenetic molecules and intraperitoneal administration of reagents. The results of the behavioral tests performed on the saline or CNO administration groups were compared with those of the CTR and RS groups. There seemed to be indeed variabilities in baseline level (CTR group) of immobility in the TST and FST between different series of experiments (e.g., *Figure 2* vs *Figure 6*). The variabilities were likely arisen at least in part from the difference in mouse strain, C57BL/6 and PCP2-Cre;ICR mice. Mice without AAV injection were randomly divided into the CTR group and the RS group. Mice subjected to AAV injection were randomly divided into a group receiving saline administration and a group receiving CNO administration, both of which received RS 30 min after saline or CNO administration, unless stated otherwise. To verify the absence of any effects of saline or CNO administration alone on RS-dependent and RS-independent behaviors, mice without AAV injection were divided into the four groups: receiving saline (RS-Sal) or CNO (RS-CNO) during RS (*Figure 2G–I*), or receiving saline (Sal) or CNO (CNO) without RS (*Figure 7—figure supplement 1B–D*). The control experiments using six groups, that is CTR, RS, RS-Sal, RS-CNO, Sal, and CNO groups, were concurrently performed. In experiments shown in *Figure 7*, CNO or saline was administered once a day for 2 weeks without RS. Five to eight mice were used for each group in a set of concurrently processed behavioral experiments including four or six groups, and this set of experiments was repeated three to six times for a type of tests. Body weights were measured before and on the last day of 2 weeks of RS application or CNO administration, and changes in body weight during the 2 weeks were calculated. All mice receiving RS showed a significant lack in body weight gain (*Figure 2—figure supplement 2*, *Figure 6—figure supplement 1*, *Figure 7—figure supplement 2*), confirming that these mice were stressed, because a decrease in body weight gain is generally accepted as reflecting stress. (*Jeong et al., 2013*; *Filaretova et al., 2013*; *Sántha et al., 2015*). Nevertheless, changes in body weight were not affected by manipulation

of crus I PCs (*Figure 2—figure supplement 2*) or VTA-projecting DCN neurons upon CNO administration (*Figure 6—figure supplement 1*, *Figure 7—figure supplement 2*).

For chemogenetic manipulation in vivo, the CNO stock solution dissolved in dimethyl sulfoxide (100 mM) was further diluted in saline for the intraperitoneal injection (100 μL/mouse) at 10 mg/kg. We used relatively high concentrations of CNO to manipulate circuit activity for longer periods upon single injection. Although acute off-target effects of CNO were reported (*Manvich et al., 2018*; *Jendryka et al., 2019*), we confirmed in control experiments without AAV injection that there were no off-target effects in our behavioral analyses (*Figure 2G–I*, *Figure 7—figure supplement 1B–D*). For the control, only saline (without CNO) was administered. The administration of CNO or saline was carried out 30 min before applying the RS, or once a day without RS, every day for 2 weeks. As both unilateral and bilateral excitation of crus I PCs resulted in a reduction in stress-mediated depression-like behaviors, as shown in *Figure 2—figure supplement 1A–F*, combined results are presented in the main figure (*Figure 2J–L*).

## Behavioral analysis

The main symptoms of depression include feelings of worthlessness and helplessness. To evaluate depression-like behaviors, two behavioral tests, that is the TST and FST, which are widely applied to assessment of depression-like behaviors in rodent model, were primarily used. After AAV injections at P35, the mice were left for another 3–4 weeks to ensure full expression of the constructs, and then the handling process (5 min for 3 days) was started when the mice were 8–9 weeks old. After the 2 weeks of RS and/or CNO administration, three behavioral tests were performed in the next three consecutive days, that is the OFT for testing general locomotor activity, and the TST and FST for assessing depression-like behaviors. In separate sets of behavioral experiments, depression-like behaviors were assessed by the SST and NSFT, which are tests for a lack of self-care and hyponeophagia or anhedonia, respectively, performed in the next two consecutive days after the last day of RS or chemogenetic manipulation. In addition, to test whether chemogenetic activation of VTA-projecting DCN neurons could result in persistent depression-like behaviors, the OFT, TST, and FST were performed a week after the SST and NSFT. The order of behavioral tests was OFT, TST, and FST, or SST and NSFT. Although we cannot deny a possibility that the order of test may affect behavioral consequences, such as relatively smaller effects of RS in TST than FST, depression-like behaviors were always confirmed by including the CTR group and the RS group in one series of concurrently processed behavioral experiments. The results of the behavioral tests were analyzed using Ethovision software (Noldus) with an immobility threshold of 5% for the results of the TST and FST.

### Tail suspension test (TST)
Each mouse was suspended with a 20-cm-long tape from a rod, which was horizontally placed 50 cm above the floor. A 3-cm-long 15 mL conical tube was placed through the tail to prevent the mice from climbing back. The TST lasted for 6 min and video recordings of the last 4 min were used for the analysis. Immobility was considered as a mouse being completely motionless while being hung.

### Forced swimming test (FST)
Each mouse was put into an acrylic cylinder (30 cm high and 15 cm in diameter) filled with water at 22°C to 24 °C. The depth of the water was up to 15 cm of the cylinder. Trials were video-recorded for a total of 6 min, and the last 4 min of the recording were used for analysis. Immobility was considered as remaining motionless except for movements that were necessary for the mice to float and to keep their balance or keep their head/nose above the water.

### Open-field test (OFT)
The OFT was performed by placing each mouse in the center of an open field chamber, which was made of white plexiglass (30 cm × 30 cm × 30 cm). Video recording began immediately after placing a mouse in the chamber, and lasted for 30 min. The distance traveled in the horizontal plane for 30 min was measured to ensure that the immobility in the TST and FST was not due to a reduction in general locomotion. To prevent anxiety-like behaviors affecting results of the OFT, we performed the OFT in relatively small chamber without bright light illumination. For the data used in *Figure 1D*, time spent in center and peripheral areas were also analyzed, by dividing the box into 16 square zones using the

Ethovision software (*Figure 1—figure supplement 1B*). The middle four square zones were considered to be the center whereas the rest of the square zones were counted as peripheral areas.

## Sucrose splash test (SST)

The SST was performed in the home cage with wooden bedding. A mouse was placed in the cage, and sprayed with approximately 200 µl sucrose solutions (10%) directly onto its back. Grooming behavior was then recorded for 5 min, and total duration of grooming was measured. Touching, scrubbing and licking their fur was considered as the grooming behaviors.

## Novelty suppressed feeding test (NSFT)

The NSFT was performed by scoring the latency to feed, when a food-deprived mouse is introduced to an unfamiliar environment in the white plexiglass container (50 cm × 50 cm × 35 cm) with the floor covered by wooden bedding. All food was removed 24 hr prior the test in the home cage. A single sweet food pellet (soaked in 50% sucrose) was placed in the center of the container that was brightly illuminated (500 lux). Once a mouse was placed in the test container, the latency to feed was measured during 10 min. Mice that exceeded 10 min without eating the pellet were excluded from the data analysis. After the test, the mouse was immediately returned into its home cage and further recorded for 5 min to measure latency to feed in the home cage as a sign of control feeding drive, which was not significantly different between groups (*Figure 6G*, *Figure 7—figure supplement 1H*).

## Statistical analysis

All statistical tests were performed using Origin Pro software. Behavioral data are presented as boxplots with gray dots representing individual data points, center lines denoting the median, open square dots denoting the mean values, the lower and the upper bounds of the box corresponding to the 25th and 75th percentiles, respectively, and the whiskers denoting the minimum and maximum values. Other data are presented as the mean ± SEM. The unpaired Student's $t$-test was used for two-group comparisons, and one-way ANOVA followed by the Fisher's least significant difference (LSD) post hoc test was conducted to compare significant differences between more than two groups. We described summary statistics in Results, and listed details of statistical information, including exact p values and the statistical tests used, in all source data one related to *Figure 1—source data 1*, *Figure 2—source data 1*, *Figure 4—source data 1*, *Figure 5—source data 1*, *Figure 6—source data 1*, *Figure 7—source data 1*. To explore the possibility that individual variability in the TST and FST would be in part due to the degree of depression-like behaviors, or due to the degree of abnormal locomotion, correlation between TST and FST, TST and OFT, or FST and OFT was examined using Pearson's correlation coefficient (*Table 1*). A $p$ value of less than 0.05 was considered to indicate a statistically significant difference between groups, and the sample numbers in individual groups are presented in the figure legends. Effect sizes of behavioral treatments estimated by using Cohen's d ( = (M1 - M2) / average SD) were summarized in all source data two related to *Figure 2—source data 2*, *Figure 6—source data 2*, *Figure 7—source data 2*. Numbers of animals used for behavior analyses were not statistically predetermined but conform to similar studies (e.g. *Carta et al., 2019*; *Kelly et al., 2020*).

## Acknowledgements

We thank Dr. Taegon Kim, Dr. Heeyoun Park, Ms. Soyoung Jun, Ms. Seul gi Kang, Mr. Muwoong Kim, and Ms. Gina Shim for valuable discussions during the project and for their technical support, and Dr. Helena Akiko Popiel for proofreading the manuscript. This work was supported by the National Research Foundation of Korea (NRF) Grant funded by the Korean Ministry of Science and ICT (MSIT) (NRF grant no. 2016R1A2B3008165 and 2021R1A2C3009991), the KIST Institutional Program (project no., 2E30971), and National R&D Program through the NRF funded by MSIT (2021M3F3A2A01037808).

# Additional information

## Funding

| Funder | Grant reference number | Author |
|---|---|---|
| National Research Foundation of Korea | National Research Foundation of Korea (NRF) Grant | Soo Ji Baek<br>Keiko Tanaka-Yamamoto |
| Korea Institute of Science and Technology | KIST Institutional Program | Soo Ji Baek<br>Jin Sung Park<br>Jinhyun Kim<br>Yukio Yamamoto<br>Keiko Tanaka-Yamamoto |
| National Research Foundation of Korea | National R&D Program | Keiko Tanaka-Yamamoto |

The funders had no role in study design, data collection and interpretation, or the decision to submit the work for publication.

## Author contributions

Soo Ji Baek, Data curation, Formal analysis, Writing - original draft, Writing - review and editing; Jin Sung Park, Data curation; Jinhyun Kim, Yukio Yamamoto, Conceptualization, Supervision, Writing - review and editing; Keiko Tanaka-Yamamoto, Conceptualization, Funding acquisition, Supervision, Writing - original draft, Writing - review and editing

## Author ORCIDs

Yukio Yamamoto (ID) http://orcid.org/0000-0002-8530-0701
Keiko Tanaka-Yamamoto (ID) http://orcid.org/0000-0002-1246-9784

## Ethics

All experiments were performed in accordance with the Institutional Animal Care and Use Committee of Korea Institute of Science and Technology (KIST-2021-11-145).

## Decision letter and Author response

Decision letter https://doi.org/10.7554/eLife.72981.sa1
Author response https://doi.org/10.7554/eLife.72981.sa2

# Additional files

## Supplementary files

• Transparent reporting form

## Data availability

All data generated or analysed during this study are included in the manuscript and supporting file; Source Data files have been provided for statistics and effect sizes.

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
