## [Editor Report]

This study provided novel insight into a role of a cerebellar-ventral tegmental area (VTA) circuit that is recruited by chronic stress exposure and can influence affective behavior. A combination of experimental manipulations, carefully performed and analyzed, dissecting the cerebello-VTA circuit with neuroanatomy, c-fos expression, behavior, and chemogenetics, supports the conclusion of the paper, that the VTA-projecting cerebellar neurons proactively regulate the development of stress-dependent changes in affective behavior. This report provides convergent evidence for a novel pathway that modulates chronic stress-related behaviors, and makes an important advance in the field.

---

## [Decision Letter]

**Decision letter after peer review:**

Thank you for submitting your article "VTA-projecting cerebellar neurons mediate stress-dependent depression-like behavior" for consideration by *eLife*. Your article has been reviewed by 3 peer reviewers, one of whom is a member of our Board of Reviewing Editors, and the evaluation has been overseen by Kate Wassum as the Senior Editor. The reviewers have opted to remain anonymous.

Essential revisions:

1) The reviewers all agree that there is only one set of additional control studies required. The interpretation that CNO does not have off target effects because it doesn't have an impact in non-stressed animals is not a valid control, as CNO could still possess off target effects that are only relevant under conditions of stress (for example through its ability to interact with serotonin receptors), so an appropriate control required would be to test CNO under stress conditions in the absence of the DREADD constructs and ensure it has no influence. As such, we would require the authors to demonstrate that CNO in the absence of the DREADD construct has no influence on the behavioral tests, in a similar manner to what they have already demonstrated in non-stressed animals.

2) In addition to this one experimental study that needs to be added, there are several additional issues that are required to be addressed by the authors but these can be done through clarification and revision of the manuscript itself.

a. The authors use a lot of language that too strongly suggests direct translation of these data to human conditions, such as referring to the behavioral state as "mental depression" and such. All of the language regarding this anthropomorphization of depression must be removed and clarified. It is acceptable for the authors to discuss the potential translational relevance of these findings in the discussion in the context of depression and how these data may apply, but to state that the mice are "depressed" or in a state of "mental depression" after stress exposure is not appropriate. Please modify all of the language used in this capacity throughout the manuscript.

b. The timing of the CNO administration relative to the stress exposure and the behavioral testing indicates that silencing this cerebellar-VTA pathway during stress can prevent the development of stress-induced behavioral changes, but since the authors did not perform the additional experiment of doing chronic stress in the absence of CNO and then administering CNO only before the behavioral tests were performed, they cannot claim that this pathway is involved in the expression of these behaviors. If the authors would like to be able to make this claim they would need to do this additional study, but in the absence of adding this study, the language in the paper must be modified so that its clear that their data only apply to the development of these behavioral changes and not their expression.

c. There are several points regarding potential interactions among local cerebellar circuitry that need to be addressed.

i. The authors do not explain the significance that VTA-projecting dentate nucleus neurons have reciprocal connections with Purkinje cells. Couldn't this in fact provide an alternative mechanism accounting for how DREADD manipulations of DCN-VTA pathway neurons could in fact affect cerebellar processing writ large? Please clarify.

ii. GFP signal of the projections from the DCN to the midbrain is shown in Figure 1 and SuppFig1 and it is visible that most of the projections go to red nucleus (RN) instead of VTA. There are also projections to the ipsilateral side in the posterior VTA. Please discuss the function of RN and how could it be related to the findings of this paper, and also the possible function of the ipsilateral VTA projections.

iii. Axon collaterals of DCN-VTA projection neurons to the cerebellar cortex are presented in Figure 3. Blue and green co-localization in VTA is not very clear (colors very similar); also, please provide an image with GFP in the whole cerebellar cortex with the mossy fiber terminals expressing GFP to see if other areas than crus 1 are showing similar patterns.

iv. After chemogenetic stimulation over 2 weeks of crus I presented in Figure 5, c-fos levels are as low as in the control group that only received stress. The authors conclude this makes this setup an appropriate experimental setup for testing the involvement of the cerebellum in development of depressive symptoms. May chemogenetic stimulation of crus I during the stressful event over 2 weeks have induced plasticity also in other deep parts of the cerebellum that receive projections from crus I?

v. DN activity was inhibited chemogenetically during chronic stress application in Figure 6. Part of the effect might be due to inhibition of neurons that are anatomically close to the DN but also form synapses with VTA? (e.g. IPN) please discuss.

d. c-fos expression is presented over 10 days; could you present the percentage of cfos-positive neurons increased over time? c-Fos was co-stained with dapi, do you know what kind of cells were stained? In addition to this analysis the authors should also present their fos data in terms of numbers not percentages so that this can be seen as well.

e. As the authors have noted, others targeting this pathway have found opposite results to what has been presented here (i.e., stimulation of this pathway produces reward not "depression"), in the Carta et al., 2019 reference. Further discussion of the methodological differences between these studies, especially the difference in both timing (manipulating the circuit during the behavior itself as opposed to during the stress exposure) and approach (optogenetics vs chemogenetics) is necessary.

f. The behavioral analysis in the open field test of only measuring motor activity is not directly related to affective behavior as this is usually measured by assessing how much time the mice spend in the central quadrant of the arena versus the peripheral areas. Please reanalyze this data to include measures of time spent in the center vs. periphery.

*Reviewer #1:*

The authors examined the role of a cerebellar-VTA circuit that contributed to stress-induced behavioral changes.

Strengths:

–The authors use a wide array of techniques and create a compelling data set using anatomy, physiology and multiple intervention approaches to demonstrate that this cerebellar-VTA circuit is recruited during chronic stress and is both sufficient and necessary for behavioral changes from stress that could have relevance for depression.

– In many cases the authors use redundant approaches to demonstrate an outcome, which provides confidence in these measures.

– The circuit the authors elucidate is very novel for its potential role in affective behavior and will likely help to enhance consideration of a putative role of the cerebellum in affective illnesses

Weaknesses:

– The authors describe their behavioral end points too casually as depression, the wording should be changed to reflect that these are simply stress-induced behavioral changes and save any mention of depression to the discussion where it is couched appropriately. To refer to these behaviors as helplessness or despair is anachronistic and not appropriate anymore.

– The authors interchange the use of chronic restraint with chronic mild stress, yet these paradigms are fundamentally different and produce a wide range of different outcomes due to the fact that chronic restraint is a habituating stressor and chronic mild stress is not. The language around this should be clarified.

– The interpretation that CNO does not have off target effects because it doesn't have an impact in non-stressed animals is not a valid control, as CNO could still possess off target effects that are only relevant under conditions of stress, so an appropriate control required would be to test CNO under stress conditions in the absence of the Dreadd constructs.

– The paper would be strengthened by having a clear understanding of the impact of changes in this cerebellar circuit on VTA neurons under conditions of chronic stress as while the cerebellar arm of the circuit has been manipulated, its not entirely clear how this influenced stress-induced changes in VTA function.

Comments for the authors:

From an experimental standpoint, the inclusion of a CNO control under conditions of stress is a necessary, but relatively easy, addition to make. This would provide confidence in the outcome measures.

From a manuscript standpoint, a lot of the language regarding what the behavioral tests are measuring needs to be tempered. There is far too much antropomorphization about mental state, depressive symptoms and which comples domains these simple behavioral tests are tapping into and so this needs to be significantly edited.

*Reviewer #2:*

The manuscript by Baek et al. mechanistically examines the role of a novel pathway involving a deep cerebellar nuclear (DCN) projection to the ventral tegmental area (VTA) using a battery of tests regarded as "depression-like." These studies show that chronic chemogenetic inhibition of the DCN-VTA pathway throughout 14 d of daily restraint stress prevented subsequent alterations in forced swim, tail suspension, novelty suppressed feeding, and sucrose splash tests. That the authors further showed that chemogenetic activation of this pathway in the absence of stress promotes chronic stress-like changes in these behaviors lends to their overall conclusion that DCN input to VTA, on its own, may be capable of driving a chronic stress-like state. This is a very comprehensive study, and contains a progression of carefully designed experiments that provide convergent support for a novel circuit involvement in stress-related behavioral changes. In their current form, these data are sufficient to provide an interesting and very useful translationally-relevant narrative, however there are a number of critical issues regarding the mechanistic basis underlying the authors' claims. These along with several other comments will be highlighted below:

Time course and mechanism of circuit effects:

The DREADD manipulations involved daily administration of CNO, given daily before each restraint episode over a 2-week period. After the end of the 2-week restraint stress regimen, rats were tested on subsequent days in the behavioral assays. The controls convincingly demonstrated that the chronic DREADD manipulations of the DCN-VTA pathway reliably alter the behavioral endpoints of interest vis-à-vis other potential confounds of the CNO itself. However, the mechanism accounting for circuit involvement in these effects remains obscure given the timing of CNO administration prior to each daily stress episode. Confusion surrounding this issue is furthered by the facts that (1) Fos induction in DCN appears to be non-specific regardless of when the mouse undergoes restraint (i.e., 3, 7, 10 vs. 14 d; also after a single exposure, e.g., Cullinan et al., 1995); (2) other recent evidence suggests that activating the DCN-VTA pathway in real-time (Carta et al., 2019) has effects opposite in sign to the authors' findings.

Regarding the first point: if DCN is encoding some aspect of the stressor in order to yield depression-like behavior as the authors propose, then this activity should change (e.g., increase) after some period of time or otherwise display temporal characteristics consistent with this idea. At least, it would be reasonable to expect that increases in Fos in this pathway accompany depression-like behavioral changes during the specific behavioral test, e.g., FST. Instead, since the CNO manipulations are made chronically and prior to restraint each day, and neither Fos nor any other type of activity is assayed after any behavioral endpoint, it is impossible to directly associate Fos or DREADD circuit with behavioral changes.

Second, recent evidence shows that optogenetically activating the DCN-VTA pathway in mice is both rewarding and increases social interaction. Each of these effects are opposite to the behavioral phenotypes observed following chronic stress. Granted, the onus needn't be on the authors to account for every discrepancy between their and past studies. However, given the aforementioned concerns about the time course and chronicity of CNO manipulations in the current study, and uncertainty over what effects these manipulations have over the circuit, the precedent from Carta et al. further destabilizes the authors' central interpretation.

Reciprocal connectivity:

The authors do not explain the significance that VTA-projecting dentate nucleus neurons have reciprocal connections with Purkinje cells. Couldn't this in fact provide a alternative mechanism accounting for how DREADD manipulations of DCN-VTA pathway neurons could in fact affect cerebellar processing writ large? Maybe this sorts out with better explanation from the authors.

Controls for DREADD experiments:

I appreciate that the authors have taken pains to show that there are no effects of vehicle-only and CNO-only manipulations on the behaviors of interest. However, based on this information, the authors then choose the wrong control group for all subsequent experiments- the saline control. The correct control in DREADD studies is a CNO-only control. That the authors judge that the lack of using the correct control group is reasonable; nevertheless it remains a sub-optimal design.

*Reviewer #3:*

In this manuscript the authors investigated the role of the cerebello- ventral tegmental area (VTA) circuit in chronic stress-induced development of depression-like behavior. To address this question, the authors first studied a neuronal circuit from crus I of the cerebellar hemisphere to the VTA through the dentate nucleus of the cerebellar nuclei. Then they examined the development of depression-like behavior during chronic stress application and the involvement of cerebellum in this behavior using c-fos expression and of VTA-projecting cerebellar neurons using chemogenetics. They concluded that VTA-projecting cerebellar neurons mediate stress-dependent depression-like behaviour. Overall, this is a very nice piece of work. Nevertheless, I have some comments below that I feel will help improve the clarity of the paper.

1. GFP signal of the projections from the DCN to the midbrain is shown in Figure 1 and SuppFig1 and it is visible that most of the projections go to red nucleus (RN) instead of VTA. There are also projections to the ipsilateral side in the posterior VTA. Please discuss the function of RN and how could it be related to the findings of this paper, and also the possible function of the ipsilateral VTA projections.

2. The authors identified neurons that react to light stimulation on GFP-Chr2 positive axons by patch clamping neurons in VTA slices in Figure 2. Both putative DA neurons and GABAergic neurons were observed in VTA. Where these neurons located in the same place or close to the RN?

3. Axon collaterals of DCN-VTA projection neurons to the cerebellar cortex are presented in Figure 3. Blue and green co-localization in VTA is not very clear (colors very similar); also, please provide an image with GFP in the whole cerebellar cortex with the mossy fiber terminals expressing GFP to see if other areas than crus 1 are showing similar patterns.

4. Authors analyzed immobility as a measure of depression; this might be inaccurate, as the immobility could also be due to exhaustion.

5. c-fos expression is presented over 10 days; could you present the percentage of cfos-positive neurons increased over time? c-Fos was co-stained with dapi, do you know what kind of cells were stained?

6. After chemogenetic stimulation over 2 weeks of crus I presented in Figure 5, c-fos levels are as low as in the control group that only received stress. The authors conclude this makes this setup an appropriate experimental setup for testing the involvement of the cerebellum in development of depressive symptoms. May chemogenetic stimulation of crus I during the stressful event over 2 weeks have induced plasticity also in other deep parts of the cerebellum that receive projections from crus I?

7. DN activity was inhibited chemogenetically during chronic stress application in Figure 6. Part of the effect might be due to inhibition of neurons that are anatomically close to the DN but also form synapses with VTA? (e.g. IPN) please discuss.

8. The immobility level is lower in mice with gi/CNO injections than saline injections and similar to control group levels that did not experience stress.

9. Excitation of DCN-VTA projection neurons using Gq chemogenetic activation leads to depression like behaviour in the context of chronic stress (Figure 7). The group that received Gq-CNO has reduced exploring activity compared to the chronically stressed group that did not receive anything else. What could be the reason?

10. The authors performed chemogenetic manipulations. Carta et al. have shown in 2019 that DCN to VTA projections are rewarding instead of inducing depression. Please discuss the interest in the use of optogenetics to study this pathway during chronic restraint stress. Could chemogenetic manipulations affect short term memory and the reaction to a stressful situation?

---

## [Author Response]

Essential revisions:1) The reviewers all agree that there is only one set of additional control studies required. The interpretation that CNO does not have off target effects because it doesn't have an impact in non-stressed animals is not a valid control, as CNO could still possess off target effects that are only relevant under conditions of stress (for example through its ability to interact with serotonin receptors), so an appropriate control required would be to test CNO under stress conditions in the absence of the DREADD constructs and ensure it has no influence. As such, we would require the authors to demonstrate that CNO in the absence of the DREADD construct has no influence on the behavioral tests, in a similar manner to what they have already demonstrated in non-stressed animals.

We appreciate all the reviewers agreeing that there is only one set of additional experiments required.

We also thought that testing the impact of CNO in stressed mice, without chemogenetic molecule expression, is an important control experiment, in order to claim that CNO does not have off-target effects on stress-dependent behavioral changes. We indeed have performed this control experiment, in which we found that administration of CNO did not affect stress-mediated behavioral changes, and have included the results as supplemental figures (Figure 5—figure supplement 1A–C) of the original manuscript. We assume that the reason why the reviewers missed the results would be because they were in the supplemental figure, despite their importance. In addition, we admit that our original sentence after the test of CNO effects on non-stressed animals (Figure 7—figure supplement 1B-D) was vague, and it may have been confusing.

In the revised manuscript, we therefore presented the results of CNO effects on stressed animals in the main figure (Figure 5G–I), and modified the descriptions in the Results to emphasize them (lines 356 – 369 in Word; lines 362 – 375 in PDF). To clarify that the results shown in Figure 7—figure supplement 1B-1D are to confirm the absence of off-target effects of CNO on behaviors of non-stressed animals, we added a description “without RS application” in the sentence (line 473 in Word; line 481 in PDF).

2) In addition to this one experimental study that needs to be added, there are several additional issues that are required to be addressed by the authors but these can be done through clarification and revision of the manuscript itself.a. The authors use a lot of language that too strongly suggests direct translation of these data to human conditions, such as referring to the behavioral state as "mental depression" and such. All of the language regarding this anthropomorphization of depression must be removed and clarified. It is acceptable for the authors to discuss the potential translational relevance of these findings in the discussion in the context of depression and how these data may apply, but to state that the mice are "depressed" or in a state of "mental depression" after stress exposure is not appropriate. Please modify all of the language used in this capacity throughout the manuscript.

We apologize for our inappropriate way of expression. We should have carefully chosen the terms when we describe behavioral changes triggered by the stress application in mice.

The review articles discussing rodent models of depression have used terms such as depression-like behaviors/states in rodents (Wang et al., Prog. Neuropsychopharmacol. Biol. Psychiatry, 2017; Planchez et al., J. Neural Transm., 2019; Gururajan et al., Nat. Rev. Neurosci., 2019).

In the revised manuscript, we therefore used “depression-like behaviors/symptoms” when we describe several types of behavioral changes triggered by stress in mice throughout the manuscript, and stated the definition of the terms in the Results (lines 276 – 282 in Word; lines 280 – 286 in PDF). On the other hand, we used “depressive symptoms/disorder” or “mental depression”, only when we refer symptoms in human, and when we discuss the relevance of our experimental findings in mice in the context of depression, as suggested by the reviewers.

b. The timing of the CNO administration relative to the stress exposure and the behavioral testing indicates that silencing this cerebellar-VTA pathway during stress can prevent the development of stress-induced behavioral changes, but since the authors did not perform the additional experiment of doing chronic stress in the absence of CNO and then administering CNO only before the behavioral tests were performed, they cannot claim that this pathway is involved in the expression of these behaviors. If the authors would like to be able to make this claim they would need to do this additional study, but in the absence of adding this study, the language in the paper must be modified so that its clear that their data only apply to the development of these behavioral changes and not their expression.

We absolutely agree with this opinion. In fact, our focus in this study was to test the involvement of cerebellum and DCN neurons projecting to the VTA in the development of behavioral changes triggered by chronic stress, and we therefore used chemogenetic manipulation during chronic stress application. However, we admit that our description in the original manuscript was sometimes unclear whether neuronal pathways we tested are also involved in the expression of behavioral changes, as pointed out by the reviewers.

To clarify that our results only apply to the development, but not the expression, we added and modified descriptions in the discussion regarding our focus on the development of stress-induced behavioral changes (lines 575 – 586 in Word; lines 587 – 597 in PDF). In addition, we included “development” in parts summarizing our study (lines 31 and 540 in Word; lines 32 and 551 in PDF), and modified a sentence in the Abstract to specify our “chronic” chemogenetic manipulation (lines 27 – 30 in Word; lines 27 – 30 in PDF).

c. There are several points regarding potential interactions among local cerebellar circuitry that need to be addressed.i. The authors do not explain the significance that VTA-projecting dentate nucleus neurons have reciprocal connections with Purkinje cells. Couldn't this in fact provide an alternative mechanism accounting for how DREADD manipulations of DCN-VTA pathway neurons could in fact affect cerebellar processing writ large? Please clarify.

Since we showed that the VTA-projecting DCN neurons had the nucleocortical pathway that projects back from the DCN to the cerebellar cortex, we reaffirm the importance of discussing contributions of this pathway to the behavioral changes we observed.

Considering a previous study of associative motor learning that demonstrated contributions of nucleocortical projections to the amplification of learned response (Gao et al., 2016), it is possible that the nucleocortical pathway of VTA-projecting DCN neurons also contributes to the stress-dependent behavioral changes. However, this previous study showed that the optogenetic activation of nucleocortical mossy fibers did not trigger the learned response in untrained animals, while it increased the learned response in trained animals. If such modulatory effects are common in the nucleocortical pathway, stressdependent behavioral changes triggered by chronic excitation of VTA-projecting DCN neurons alone would not be only due to the activation of nucleocortical pathway, but possibly due to the nucleocortical pathway amplifying responses triggered by the efferent pathway.

In our original manuscript, we had a short discussion about the involvement of nucleocortical pathway in the development of stress-dependent behavioral changes, by referring the study of Gao et al. (2016).

In the revised manuscript, we expanded the discussion as mentioned above, and transferred to the part where we discussed about axon collateral pathways of DCN neurons (lines 614 – 625 in Word; lines 626 – 637 in PDF).

ii. GFP signal of the projections from the DCN to the midbrain is shown in Figure 1 and SuppFig1 and it is visible that most of the projections go to red nucleus (RN) instead of VTA. There are also projections to the ipsilateral side in the posterior VTA. Please discuss the function of RN and how could it be related to the findings of this paper, and also the possible function of the ipsilateral VTA projections.

The red nucleus is indeed one of major projection areas of DCN neurons, especially in the IPN and DN. Therefore, when DCN neurons in the IPN or DN are non-selectively labeled as shown in Figure 1A–C and Figure 1—figure supplement 1, large amounts of axons in the RN can be visible. Importantly, when DCN neurons projecting to the VTA were selectively labeled, the GFP signals in the RN were considerably reduced (Figure 1G and Figure 3B). However, because GFP signals were still weakly visible in the RN even after selective labeling, collateral axons of VTA-projecting DCN neurons may partly project to the RN, and we admit that possible involvement of projections to the RN in our findings cannot be denied.

In terms of functions of projections to the RN, the cerebellar-RN pathway is generally considered to have motor functions. Specifically, a pathway from the IPN to the RN is well known for associative motor learning, namely, delay eyeblink conditioning. Evolutionarily, RN is divided into two different parts, magnocellular (mRN) part and parvocellular (pRN) part, latter of which is larger and more clearly visible in primate or human than relatively primitive animals. This raises a possibility that pRN could be involved in cognitive functions. While the IPN mainly projects to the mRN, which is a projection responsible for the delay eyeblink conditioning, the DN mainly projects to the pRN (Basile et al., Brain Struct. Funct., 2021). Therefore, it could be speculated that the DN-pRN projections may contribute to cognitive functions. Nevertheless, because the relevance of the projections to stress-related behaviors is totally unknown, it is difficult to discuss further how the collateral projections to the RN is involved in the stress-dependent behavioral changes we observed here.

In the revised manuscript, we mentioned general strong projections to the RN from the DN, in addition to the IPN, in the Results part (lines 110 – 111 in Word; lines 110 – 111 in PDF). Further, we added abovementioned discussion regarding possible collateral projections to the RN and their functions in the section, where we discussed collateral projections of VTA-projecting DCN neurons (lines 625 – 632 in Word; lines 637 – 645 in PDF).

Regarding ipsilateral VTA projections, the posterior parts in Figure 1—figure supplement 1 include decussation of superior cerebellar peduncle, and GFP signals in ipsilateral side, at least in part, arise from the decussation of superior cerebellar peduncle. Because faint GFP signals appear to be partially overlapped with TH staining, ipsilateral projections from the DN to the VTA may be still possible. Although a general idea was that DCN neurons ipsilaterally project to spinal cord or brainstem, and contralaterally project to midbrain or thalamus, they don’t appear to be completely segregated (Kebschull et al., 2020; Sathyamurthy et al., 2020). Therefore, it may be reasonable, even if there are also ipsilateral VTA projections. However, considering not only our study but also other studies (Carta et al., 2019; Beier et al., 2015), it is unfortunately unclear whether DCN neurons from the DN truly form functional synapses onto neurons in the ipsilateral VTA. In addition, it is difficult to predict the function of the ipsilateral VTA projections, because it is not known how both sides of VTA cooperate each other. We therefore would like to avoid the discussion about ipsilateral VTA projections.

Nevertheless, because a possibility of ipsilateral VTA projections is remaining, we modified a description in the Results as follows (lines 110 – 113 in Word; lines 110 – 113 in PDF), so that the description would not give an impression that there are only contralateral projections.

While the injection into the DN …, it also resulted in GFP signals **mainly** in the contralateral side of both the anterior and posterior VTA, and the signals were most abundant compared with the injection into the other two nuclei.

iii. Axon collaterals of DCN-VTA projection neurons to the cerebellar cortex are presented in Figure 3. Blue and green co-localization in VTA is not very clear (colors very similar); also, please provide an image with GFP in the whole cerebellar cortex with the mossy fiber terminals expressing GFP to see if other areas than crus 1 are showing similar patterns.

In terms of blue and green signals in the VTA of Figure 3A, we added Figure 3B in the revised manuscript, and separately showed tdT (red), mTBFP (blue), and GFP (green) signals together with TH staining, so that individual signals can be seen.

The GFP signals in mossy fiber terminals of VTA-projecting DCN neurons are actually faint and sparse in low magnified images of cerebellar slices, and it is impossible to make them visible by showing the whole cerebellar slices. However, we were able to manage to find mossy fiber-like structures by enlarging them on the computer screen, and therefore could target areas for focused observations by using a high magnification objective. We then acquired enlarged images of mossy fiber terminals by z-stack imaging, and showed 3D projection images on the left of Figure 3C and in the inset of Figure 3—figure supplement 1.

In summary, although it is not possible to show mossy fiber terminals expressing GFP in low magnified images of whole cerebellar slices, we have shown enlarged 3D projections of them located not only in crus I, but also in crus II and simplex lobe, in the Figure 3—figure supplement 1A, and described in the Results (lines 219 – 227 in Word; lines 222 – 230 in PDF).

In addition, to clarify how we acquired the images of mossy fiber terminals, we modified a description in the Methods of revised manuscript (lines 833 – 836 in Word; lines 848 – 851 in PDF).

iv. After chemogenetic stimulation over 2 weeks of crus I presented in Figure 5, c-fos levels are as low as in the control group that only received stress. The authors conclude this makes this setup an appropriate experimental setup for testing the involvement of the cerebellum in development of depressive symptoms. May chemogenetic stimulation of crus I during the stressful event over 2 weeks have induced plasticity also in other deep parts of the cerebellum that receive projections from crus I?

We would like to first confirm our results. The chemogenetic stimulation over 2 weeks of crus I PCs during stress application reduced c-Fos levels as low as in the control group that “did not” receive stress. Therefore, in terms of the c-Fos level in the DN, this manipulation seems to be an experimental system applicable to test whether the increase in the c-Fos levels in the DN would be involved in triggering stress-dependent behavioral changes. However, as pointed out by the reviewers, we admit that it could be possible to trigger plasticity in other DCN subregions as well, by the current chemogenetic stimulation of crus I PCs over 2 weeks.

Therefore, we included this possibility with the description regarding possible adequacy of experimental setup (lines 342 – 345 in Word; lines 347 – 351 in PDF). We also slightly toned down the description by modifying to “our chemogenetic manipulation of PCs appeared to be an experimental system applicable to test the involvement …”, instead of saying “an appropriate experimental system” (lines 345 – 347 in Word; lines 351 – 353 in PDF).

v. DN activity was inhibited chemogenetically during chronic stress application in Figure 6. Part of the effect might be due to inhibition of neurons that are anatomically close to the DN but also form synapses with VTA? (e.g. IPN) please discuss.

We agree that the effect might be in part due to the inhibition of DCN neurons in the IPN, because we observed that the IPN also had minor projections to the VTA, and is anatomically close to the DN. We therefore had one sentence at the end of explanation of Figure 1 in our original manuscript that we were aware of the possible involvement of DCN neurons in the IPN in the subsequent experiments, even though we mainly targeted AAV injection into the DN among all DCN subregions. Nevertheless, we admit that we did not discuss or describe further about this matter in the original manuscript.

In the revised manuscript, to make this clearer, we added a possible involvement of DCN neurons in the IPN at the end of explanation of Figure 6 (lines 446 – 450 in Word; lines 454 – 458 in PDF), and also included “possibly the IPN” in summary of our study, which is the first paragraph of Discussion (lines 533 – 534 in Word; line 544 in PDF). Moreover, in the Discussion, we added a sentence to mention a possible contribution of IPN-VTA circuit to depression-like behaviors (lines 610 – 612 in Word; lines 622 – 624 in PDF).

d. c-fos expression is presented over 10 days; could you present the percentage of cfos-positive neurons increased over time? c-Fos was co-stained with dapi, do you know what kind of cells were stained? In addition to this analysis the authors should also present their fos data in terms of numbers not percentages so that this can be seen as well.

We are afraid that we may not have correctly understood the first point in this comment, but our understanding is as follows:

In our original manuscript, we showed the percentage of c-Fos-positive cells among all DAPIpositive cells in the mice receiving RS for 3, 7, 10 (Figure 4H), and 14 days (Figure 4F and 4G), as well as in the simultaneously prepared control (CTR) mice. We then found the increase in the c-Fos-positive cells by all durations of RS application, compared with corresponding CTR groups. However, the reviewers requested to also show the percentage of c-Fos-positive cells increased by RS over their CTR groups at all 3, 7, 10, and 14 days.

We therefore calculated “percentage increase in c-Fos-positive cells over CTR group” using percentages of c-Fos-positive cells in DAPI-positive cells in mice receiving x days of RS (c-Fos_RS-x_) and averaged percentages of c-Fos-positive cells in DAPI-positive cells in simultaneously analyzed control mice (c-Fos_CTR-x_): ((c-Fos_RS-x_ – c-Fos_CTR-x_)/c-Fos_CTR-x_) × 100 (%).

In the revised manuscript, we presented a graph showing results of this calculation in Figure 4—figure supplement 1E, and mentioned them in the Results (lines 311 – 314 in Word; lines 316 – 318 in PDF). We also added an explanation as to how we calculated the percentage increase in the Methods (lines 827 – 831 in Word; lines 842 – 846 in PDF).

In terms of the cell types, we have co-stained cerebellar slices obtained from mice receiving 2-week RS application, with a c-Fos antibody and either antibodies of NeuN, GAD67, or GFAP, which are markers of neurons, inhibitory neurons, and glial cells, respectively. We observed c-Fos signals overlapping with NeuN signals, but not with GAD67 or GFAP signals, suggesting that the c-Fos-positive cells are mostly glutamatergic neurons.

We additionally put images of co-stained DCN in Figure 4—figure supplement 2, and described about the cell types in the Results of revised manuscript (lines 297 – 305 in Word; lines 301 – 309 in PDF).

In addition, we added Table 2 in the revised manuscript, to show the average numbers of DAPI- and c-Fos-positive cells counted in single z-stack images, as suggested by the reviewers. The Table 2 was mentioned in the Methods (lines 831 – 832 in Word; lines 846 – 847 in PDF).

e. As the authors have noted, others targeting this pathway have found opposite results to what has been presented here (i.e., stimulation of this pathway produces reward not "depression"), in the Carta et al., 2019 reference. Further discussion of the methodological differences between these studies, especially the difference in both timing (manipulating the circuit during the behavior itself as opposed to during the stress exposure) and approach (optogenetics vs chemogenetics) is necessary.

We followed the reviewers’ suggestion, and added further discussion about the differences between our study and a study by Carta et al. (2019) in the revised manuscript (lines 659 – 673 in Word; lines 672 – 686 in PDF). Specifically, we pointed out the different approaches and temporal diversity in manipulation of the circuits, as suggested by the reviewers, and discussed an idea that the temporally different patterns of activity manipulation may have led to dissimilar consequences in these two studies.

f. The behavioral analysis in the open field test of only measuring motor activity is not directly related to affective behavior as this is usually measured by assessing how much time the mice spend in the central quadrant of the arena versus the peripheral areas. Please reanalyze this data to include measures of time spent in the center vs. periphery.

After looking carefully our original manuscript in consideration of this comment, we realized that our purpose of using the open field test (OFT) was not clear. Therefore, we feel that there may be misunderstanding, and would like to first confirm our purpose.

We certainly agree that the travel distance analyzed in OFT is not related to affective behavior, but is a measure of general locomotor activity. In fact, our primary purpose of using the OFT was to test general locomotor activity, rather than affective behavior, because it was necessary to confirm that the immobility increase observed in the TST and FST was not due to the impairment of locomotion. For this purpose, we actually tried to have less impacts of anxiety-like behavior in the OFT, because mice receiving stress sometimes tend to show reduction not only in time spent in the center, but also in travel distance in the OFT (Sturman et al., Stress, 2018; Song et al., Front. Physiol., 2020). In particular, we performed the OFT in a small chamber (30 cm × 30 cm × 30 cm) without bright light illumination, which may not be appropriate to measure affective behavior, as we mentioned in the Methods.

Regardless of our primary purpose, as suggested by the reviewers, we reanalyzed the data in the OFT used for Figure 4D, and found that the time spent in the center or peripheral areas was not significantly different between control and stress groups. Although the results could be interpreted as no anxiety-like phenotypes in our stressed mice, our results in the novelty-suppressed feeding test (NSFT) suggest that our stressed mice may actually have anxiety-like phenotypes. Therefore, no difference in time spent in the center or peripheral areas is likely because anxiety-like behavior would not be properly evaluated in our conditions of the OFT.

As mentioned above, our purpose of using the OFT was not clearly stated in our original manuscript. Therefore, in the revised manuscript, we explained our purpose of measuring general locomotor activity, when we introduced the OFT at the first time in the Results (lines 259 – 260 in Word; lines 262 – 263 in PDF). In addition, we added abovementioned results of new analysis of the time spent in the center or peripheral areas in Figure 4—figure supplement 1B, and mentioned that a possible reason of no difference between control and stressed mice might be because of the OFT conditions we used (lines 265 – 274 in Word; lines 269 – 278 in PDF). Because the conditions we used for the OFT don’t seem to be very appropriate to assess affective behavior, but we later used NSFT to assess similar behavior, we showed the time spent in the center or peripheral areas for data in Figure 4D, but not for the subsequent data.